# Rapid mechanical stimulation of inner-ear hair cells by photonic pressure

**Sanjeewa Abeytunge[1,2†], Francesco Gianoli[1†], AJ Hudspeth[2‡\*], Andrei S Kozlov[1‡\*]**

[1]Laboratory of Auditory Neuroscience and Biophysics, Department of Bioengineering, Imperial College London, London, United Kingdom; [2]Howard Hughes Medical Institute and Laboratory of Sensory Neuroscience, The Rockefeller University, New York, United States

**Abstract** Hair cells, the receptors of the inner ear, detect sounds by transducing mechanical vibrations into electrical signals. From the top surface of each hair cell protrudes a mechanical antenna, the hair bundle, which the cell uses to detect and amplify auditory stimuli, thus sharpening frequency selectivity and providing a broad dynamic range. Current methods for mechanically stimulating hair bundles are too slow to encompass the frequency range of mammalian hearing and are plagued by inconsistencies. To overcome these challenges, we have developed a method to move individual hair bundles with photonic force. This technique uses an optical fiber whose tip is tapered to a diameter of a few micrometers and endowed with a ball lens to minimize divergence of the light beam. Here we describe the fabrication, characterization, and application of this optical system and demonstrate the rapid application of photonic force to vestibular and cochlear hair cells.

**\*For correspondence:**
hudspaj@rockefeller.edu (AJH);
a.kozlov@imperial.ac.uk (ASK)

[†]These authors contributed equally to this work
[‡]These authors also contributed equally to this work

**Competing interests:** The authors declare that no competing interests exist.

## Introduction

Hair cells in the auditory and vestibular systems of vertebrates convert mechanical stimuli into electrical signals through the process of mechanoelectrical transduction (*Fettiplace and Kim, 2014*; *Martin and Hudspeth, 2021*). The mechanical receptor for such stimuli is the hair bundle, a cluster of stereocilia, or stiff enlarged microvilli, atop each hair cell. An extracellular molecular filament, the tip link, extends from the tip of each stereocilium to the side of its tallest neighbor in the plane parallel to the bundle's axis of symmetry. Mechanically gated ion channels are located at the lower end of each tip link. When a hair bundle pivots at its base toward its tall edge in response to stimulation, the increased tension in the tip links opens the ion channels and the ensuing ionic current depolarizes the cell (*Figure 1A*).

Although our understanding of the transduction process has improved significantly through the development of methods to mechanically stimulate a hair bundle, the techniques now available pose serious limitations. Two methods are commonly used to apply force to a hair bundle. The first is to deflect the bundle with a compliant glass fiber about 100 μm in length and 1 nm in diameter (*Crawford and Fettiplace, 1985*; *Howard and Ashmore, 1986*; *Howard and Hudspeth, 1988*). The fiber's tip is attached to the top of the hair bundle and its base is driven by a piezoelectric actuator. Because the preparation is immersed in an aqueous solution, however, the fiber is subjected to hydrodynamic drag that roughly doubles that on the bundle. For a typical fiber of stiffness 500 μN•m⁻¹ and drag coefficient 150 nN•s•m⁻¹, the time constant of responsiveness is about 300 μs, which corresponds to a low-pass filter with a cutoff frequency near 500 Hz (*Crawford and Fettiplace, 1985*; *Howard and Hudspeth, 1987*). Another problem is especially acute for the stimulation of mammalian hair bundles, whose stereocilia are less cohesive than those of amphibians: when a fiber is attached at a single site in the hair bundle, the displacement of other stereocilia depends in a complex manner on elastic and hydrodynamic coupling across the bundle. This arrangement

**eLife digest** The sense of hearing relies on specialized sensory cells in the inner ear. Each of these hair cells converts sounds into electrical signals that the brain can interpret. The hair cell takes its name from the bundle of rod-like structures that protrude from its top surface, which resemble hairs under the microscope. The hair bundle acts as an antenna that bends in response to sound waves. When a hair bundle moves in a particular direction, it opens ion channels in the hair-cell membrane. The resulting flow of ions into the cell triggers a cascade of events that ends with an electrical signal traveling to the brain.

Many experiments on hearing rely on being able to manipulate the movement of a hair bundle. Researchers typically use one of two methods to achieve this. In the first, a flexible glass fiber pushes against the hair bundle, whereas the second involves a jet of fluid directed against the cell. Neither of these techniques can move hair bundles fast enough for researchers to explore the vast range of sound frequencies that human ears can detect. What is more, both methods are prone to introducing errors into experiments.

Abeytunge, Gianoli et al. have developed a new method for moving hair bundles, this time with the aid of light. When light interacts with objects it exerts a photonic force. Abeytunge, Gianoli et al. show that a tapered optical fiber with a miniscule rounded lens can focus a laser beam to deliver enough photonic force to move a hair bundle. The laser beam does not damage the hair bundle, but moves it fast enough to allow researchers to study a broader range of mammalian hearing, while avoiding the errors that have bedeviled previous methods.

results in an uneven application of force to different stereocilia and can produce artifacts (*Indzhykulian et al., 2013*; *Nam et al., 2015*).

The second common method of stimulation uses a fluid jet that displaces a hair bundle through the action of a piezoelectric diaphragm (*Géléoc et al., 1997*; *Corns et al., 2014*). Although the resonant frequency of fluid injection can reach 5 kHz, practical use of the method is limited to less than 1 kHz owing to uncertainties in force calibration (*Dinklo et al., 2007*). Moreover, fluid leakage from the system might introduce a displacement bias.

In summary, the inability of current methods to reach higher frequencies by direct stimulation limits our quantitative understanding of hair-cell mechanics over more than 95% of the range of mammalian hearing, which extends to 20 kHz in humans and at least 150 kHz in some species of bats and whales. What is more, the susceptibility of current approaches to artifacts has long impeded our understanding of hearing, in particular in the case of the mammalian ear (*Nam et al., 2015*).

To address these problems, we used laser irradiation to stimulate hair bundles (*Figure 1*). Because photonic force arises when photons are absorbed, reflected, or refracted upon interaction with an object, intense illumination should apply substantial force to a bundle. Our experiments confirmed the validity of the approach and demonstrated that the requisite irradiation does not jeopardize a bundle's operation. This method allows us to probe hair-bundle physiology at previously inaccessible timescales, for the delivery time of the stimulus can accommodate the full frequency range of mammalian hearing. At the same time, this approach avoids the artifacts that bedevil current methods.

## Results

### Application of photonic force to a hair bundle

The conservation of momentum entails that reflected, absorbed, and refracted photons exert force on a target. All these phenomena are likely to take place when light strikes an array of stereocilia in a hair bundle. Although an analysis based on reflection alone would indicate that a hair bundle is relatively insensitive to radiation pressure, geometric considerations reveal that multiple modes of light propagation occur in a hair bundle by virtue of the cylindrical shape of the stereocilia (see Materials and methods). Each of these modes is capable of transferring momentum and therefore of mechanically stimulating the bundle. Because the diameter of each stereocilium compares to the wavelength

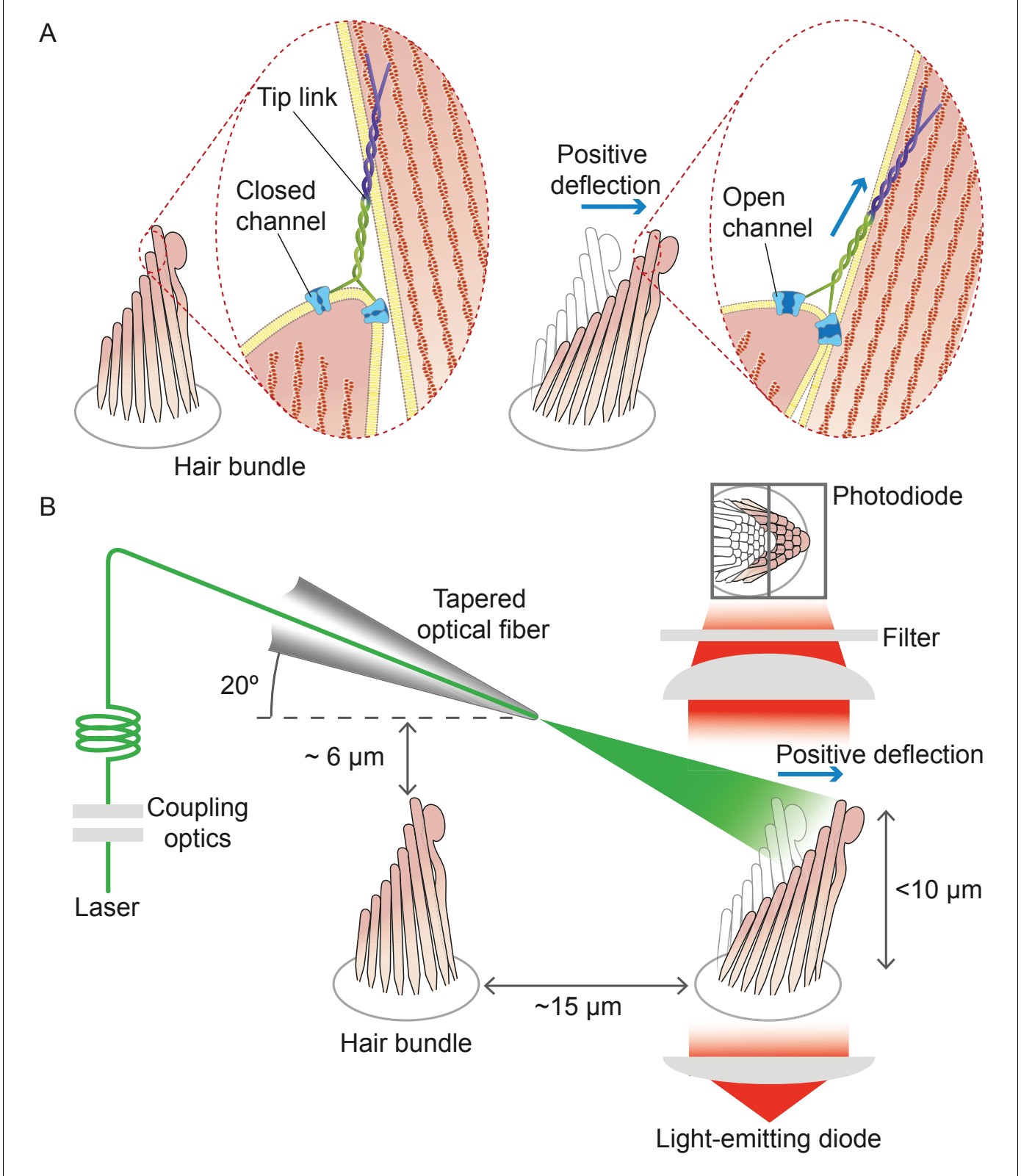

**Figure 1.** Structure of the hair bundle and configuration of the experiments. (**A**) A schematic illustration portrays a hair bundle, in this instance that from the bullfrog's sacculus, at rest (left) and when deflected towards its tall edge (right). The bundle is formed by rows of stereocilia that increase in height along the axis of sensitivity and are interlinked by molecular filaments, the tip links, that stretch as the bundle moves forward. The tip links project the stimulus force onto mechanosensitive ion channels. (**B**) A tapered optical fiber with a spherical lens at its tip is brought within a few tens of micrometers

*Figure 1 continued on next page*

*Figure 1 continued*

of a hair bundle. The fiber's angle of approximately 20 from the horizontal allows it to pass beneath the microscope's objective lens without impinging upon other nearby hair bundles. An image of the hair bundle is projected through the microscope and onto a dual photodiode, which permits measurement of bundle motion with a precision in the nanometer range. Note that the extent of hair-bundle movement in this and the subsequent figures is greatly exaggerated for didactic purposes: the largest displacements move the bundle's top by less than the diameter of a single stereocilium.

of light, the regular spacing of stereocilia within the hair bundle might additionally give rise to complex interference effects.

## Structure and orientation of an optical fiber

Geometrical factors are important in the stimulation of a hair bundle through an optical fiber. With its external plastic jacket, an intact fiber can be several millimeters in diameter. Even after the jacket has been removed, the core of the fiber—which is only 5 µm in diameter—lies within a cylinder of glass cladding about 125 µm across. In order to bring the fiber's core near a hair bundle without impingement of the fiber's outer layers on the experimental preparation, it was necessary to strip the jacket and taper the cladding. By melting the tip of the fiber's core, we created a hemispherical lens with a divergence angle in water of approximately 11° (see Materials and methods).

It was next desirable for the light beam to stimulate only a single hair bundle without affecting others nearby. This objective could be achieved readily for a flat sensory epithelium such as that of the bullfrog's sacculus, in which the bundles are about 8 µm tall and are separated by approximately 15 µm (*Figure 1*). In the rat's cochlea, however, the distance between the row of inner hair cells and the first row of outer hair cells is only 10 µm, and successive rows of outer hair cells are still more closely apposed. Moreover, this preparation is complicated by the complex curvature of its apical surface, the reticular lamina, which allows greater clearance for an optical fiber in some orientations than in others. After securing the end of a tapered optical fiber in a stable holder, we found that introducing it beneath the objective lens at an angle of 20° from the horizontal allowed the tip to approach a target hair bundle closely enough—typically about 7 µm—to ensure efficient stimulation, and at the same time positioned the tip far enough above other bundles to avoid damaging them (see Materials and methods).

## Deflection of glass rods by photonic force

Before engaging in experiments with hair bundles, we conducted control experiments to confirm that photonic force from a tapered optical fiber could move an object of stiffness comparable to that of a bundle. We thinned two glass rods with a pipette puller and measured the stiffness of each by analyzing the spectrum of its Brownian motion and applying the equipartition theorem. After positioning each rod such that its shadow projected onto the photodiode, we delivered light pulses through a tapered optical fiber positioned approximately 10 µm from the rod's tip. In both cases, we found that irradiation elicited a prompt movement in the expected direction (see *Appendix 1—figure 1A*). Having ascertained that our setup could deliver forces of an appropriate order of magnitude, we commenced experiments on living hair bundles.

## Stimulation of frog hair bundles

We stimulated 40 hair bundles of the bullfrog's sacculus so that radiation pressure would push them toward their tall edges—the positive direction—and reliably elicited the expected movements (see *Appendix 1—figure 1A*). The bundles followed similar trajectories at the onset of irradiation (*Figure 2*): the movement was approximately exponential with a time constant of $0.64 \pm 0.06\,\text{ms}$ (mean ± SEM, N = 16). The responses, which reached displacements as great as 500 nm, encompassed the range of complex trajectories reported in the literature (*Benser et al., 1996*; *Tobin et al., 2019*; *Ricci et al., 2000*). The more compliant hair bundles—those displaying initial deflections exceeding about 150 nm—displayed relatively slow movements in the direction of the photonic force, a signature of the timescale of the adaptation process that allows hair cells to reset their operating points and thus detect successive stimuli (*Figure 2A*, gray arrows) (*Howard and Hudspeth, 1987*). The twitch, a faster rebound of the hair bundle in a direction opposite to that of the stimulus, is another

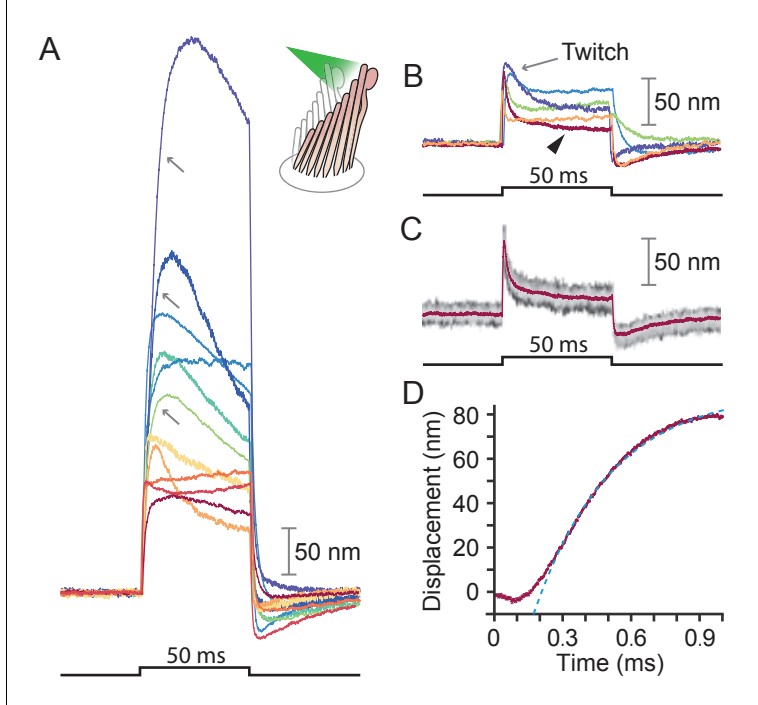

**Figure 2.** Responses of hair bundles from the bullfrog's sacculus. (**A**) Although all hair bundles displayed rapid movements at the onset and conclusion of photonic stimulation, some exhibited relatively slow approaches to their peaks and slow relaxations. Eleven hair bundles were stimulated in the positive direction with 561 nm light with 30 mW at the fiber's entrance, or about 15 mW at the fiber's output; each trace is the average of 25 responses. The schematic diagram here and in the subsequent figures shows the experimental configuration. (**B**) Five of the other hair bundles displayed moved rapidly at the onset of irradiation, then relaxed to plateau displacements. (**C**) A representative trace, marked by an arrowhead in panel B, portrays the decay of a response to a plateau level and the undershoot after stimulation characteristic of slow adaptation. The 25 individual traces that combine into the average (red) are shown in gray scale. (**D**) The rising phase of the same response is fitted with $R^2 = 0.98$ to an exponential with time constant 335 s (dashed blue line). The data at times below 250 μs were not included in the fit.

manifestation of the adaptation process that occurs instead in response to smaller movements of the hair bundle and whose magnitude decreases for larger deflections (*Benser et al., 1996*; *Ricci et al., 2000*; *Cheung and Corey, 2006*). The twitch was indeed observed in stiffer bundles with deflections of about 50 nm (*Figure 2B–C*). These results indicate that photonic force is an effective means of stimulating hair bundles.

## Polarization dependence of hair-bundle responses

Because stereocilia are densely filled with parallel actin filaments that exhibit pronounced birefringence (*Katoh et al., 1999*), we inquired how this property affected the movement of the bullfrog's hair bundles upon photonic-force stimulation. After rupturing the tip links, we imaged a hair bundle on the dual photodiode and aligned the plane of polarization with the long axis of the stereocilia. We then rotated a half-wave plate through 90° in 10° increments. The light-induced deflection declined monotonically to an angle of 40°-50°, but remained roughly constant thereafter (see *Appendix 1—figure 3*). That the response did not decline as the cosine of the angle likely reflected the fact that stereocilia are not parallel, evenly spaced cylinders but rather a more complex array with varying tilts and separations. This result nonetheless emphasized the importance of attending to the beam's polarization, which was held parallel with the hair bundles' long axes in subsequent experiments.

## Stimulation of rat hair bundles

We applied photonic stimuli to the hair bundles of both inner and outer hair cells from the cochleas of young rats. Consistent with previous evidence that mammalian hair bundles are stiffer than their amphibian counterparts (*Tobin et al., 2019*), the recorded amplitudes of deflection were typically smaller (*Figure 3A–D*). The time constants for the initial displacements were again a few hundred microseconds. To characterize the efficacy of photonic stimulation for rat hair bundles, we applied positive stimuli to 22 outer hair cells from three preparations. We deflected 13 hair bundles with

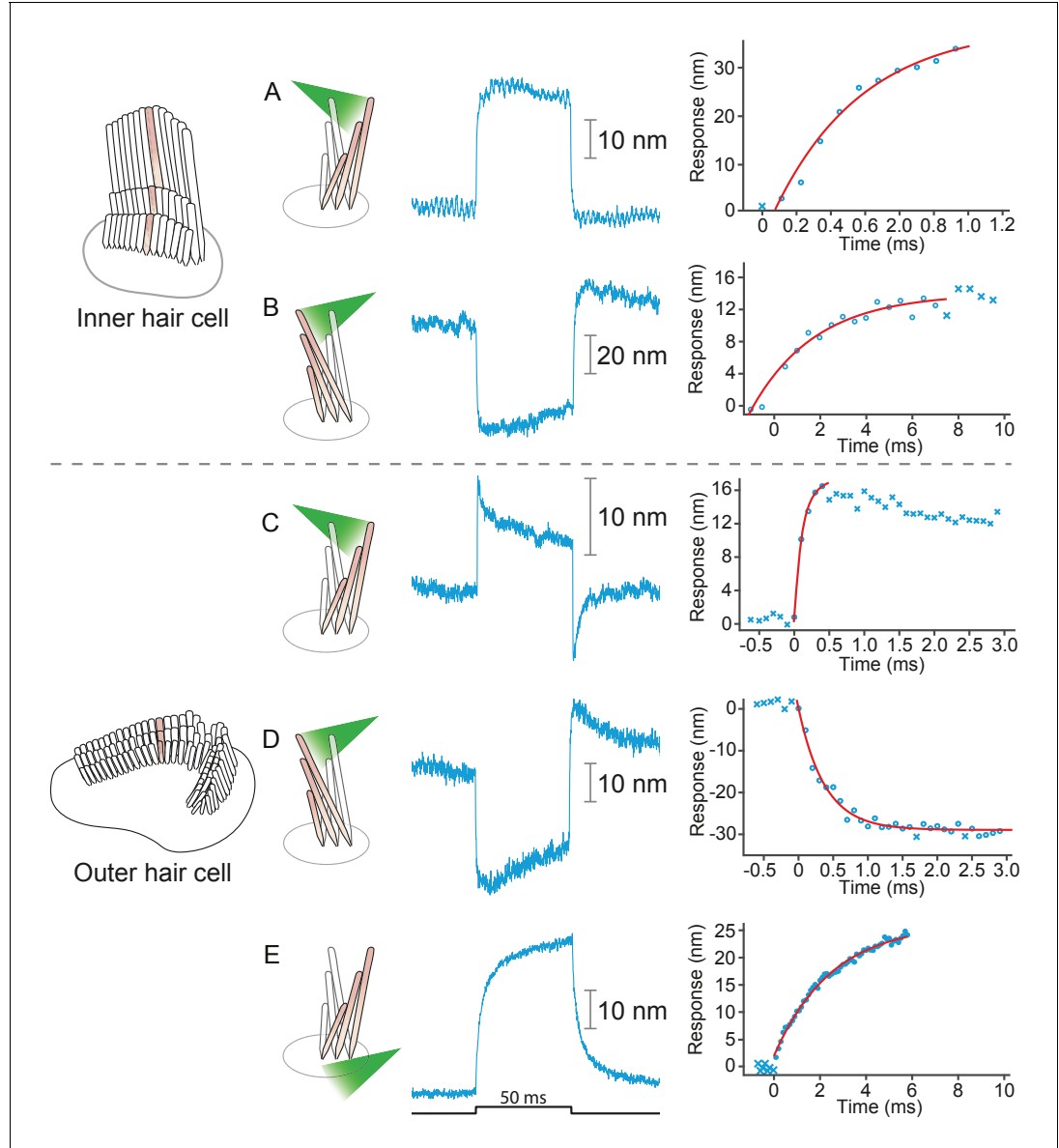

**Figure 3.** Responses of hair bundles from the rat's cochlea. (A) Irradiation of the hair bundle from an inner hair cell evoked motion in the direction of light propagation, here the positive direction, with a time constant of 459 µs. In this and the other panels, the bundles were stimulated at 600 nm with 18 mW of input power and the records represent the average of 25 repetitions. This number of repetitions was sufficient to diminish the noise and isolate the characteristic shape of the hair-bundle response. (B) A similar experiment with negatively directed irradiation moved the hair bundle in the opposite direction. The time constant is 258 µs. (C) Stimulation of an outer hair cell's bundle in the positive direction evoked a response with sharp transients at both the onset and the offset of irradiation. As shown in the associated plot, the response rose with a time constant of 123 µs and peaked in less than 1 ms. (D) Negative stimulation of an outer hair cell's bundle evoked movement in the negative direction with an onset time constant of 377 µs. (E) When a negatively directed light beam was aimed at the soma of an outer hair cell, the bundle moved with a slow time constant of 2.1 ms in the positive direction-opposite the direction of light propagation-owing to the photothermal effect.

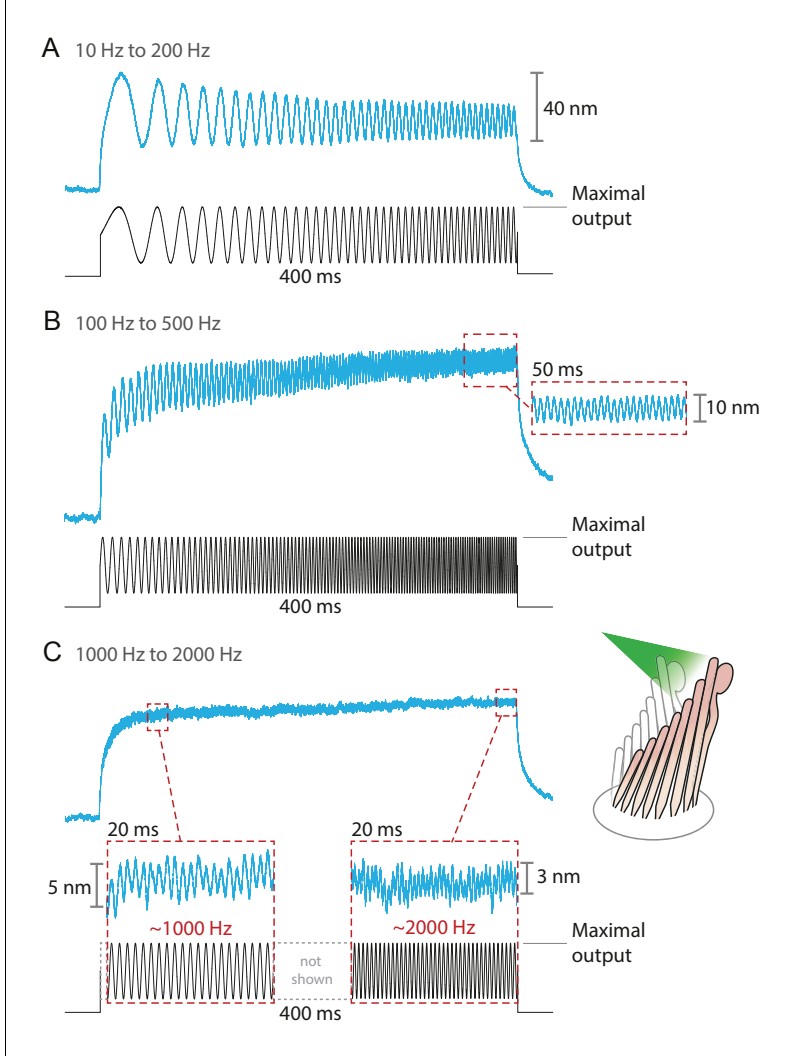

**Figure 4.** Responses of hair bundles from the bullfrog's sacculus to sinusoidal frequency sweeps between 10 Hz and 200 Hz (A), 100 Hz and 500 Hz (B), and 1 kHz and 2 kHz (C). Each stimulus was achieved by driving the laser's source such that the amplitude of the sweep peaked at the maximum power output—12.5 mW for this fiber—while keeping its minimum above 0 mW. Each hair bundle was stimulated in the positive direction with 561 nm laser light; each trace is the average of 25 responses. Panel C portrays two 20 ms-long representative segments of the stimulus waveform, which would be unintelligible if displayed in full. These segments, located near the beginning and the end of the sweep, are aligned with the magnification of the simultaneous hair-bundle response (red dashed boxes).

amplitudes varying from 25 nm to 35 nm. We also deflected seven of nine bundles from inner hair cells; the response amplitudes varied from 100 nm to 75 nm and the trajectories resembled those from the frog.

## Variety of stimuli

The fiber's power delivered onto a hair bundle can be modulated by changing the laser's power at the source. By combining analog and digital signals to drive the laser's output, we were able to stimulate hair bundles with an assortment of stimuli: sine waves, frequency sweeps, step pulses of various magnitudes, and continuously ascending and descending ramps (see *Appendix 1—figure 4*). The responses of bullfrog's hair bundles followed sinusoidal frequency sweeps at frequencies up to 2 kHz (*Figure 4*). In this case the upper boundary in the stimulus frequency was set by the ability of the hair bundle to follow, rather than by the limitations of the stimulation method.

## Separating the photothermal movement

As a result of localized heat generation, a hair bundle from the frog can move in the positive direction in response to laser irradiation of the cellular apex from any direction (*Azimzadeh et al., 2018*). We found that this phenomenon also occurs in hair bundles of the rat (*Figure 3E*). To separate this photothermal effect from that of photonic force, we took advantage of the fact that the former requires intact tip links. When we disrupted the tip links with BAPTA to abolish the photothermal effect, we observed that both positive and negative stimuli evoked rapid movements in the direction of light propagation (see *Appendix 1—figure 5*). We also stimulated hair bundles along a direction perpendicular to their axis of symmetry and again found that they moved in the direction of photon flux (see *Appendix 1—figure 5C*). These results indicate that bundle motion upon photonic stimulation can occur in the absence of a photothermal effect: bundle movements stem solely from optical radiation force.

In a frog's hair cell, the photothermal effect apparently results from light absorption by the mitochondria that accumulate around the cuticular plate at the cell's apical surface (*Azimzadeh et al., 2018*). Because in mammalian outer hair cells mitochondria are instead concentrated at the lateral plasma membrane (*Fuchs, 2010*), it was possible to isolate the photothermal effect by directing light well below the apical cell surface. Note that the photothermal movement was relatively slow: its time constant of 2 ms was about ten times that of the movements owing to photonic force. Conversely, it was possible to avert the photothermal effect by irradiating a mammalian hair bundle with intact tip links while avoiding irradiation of the cell body.

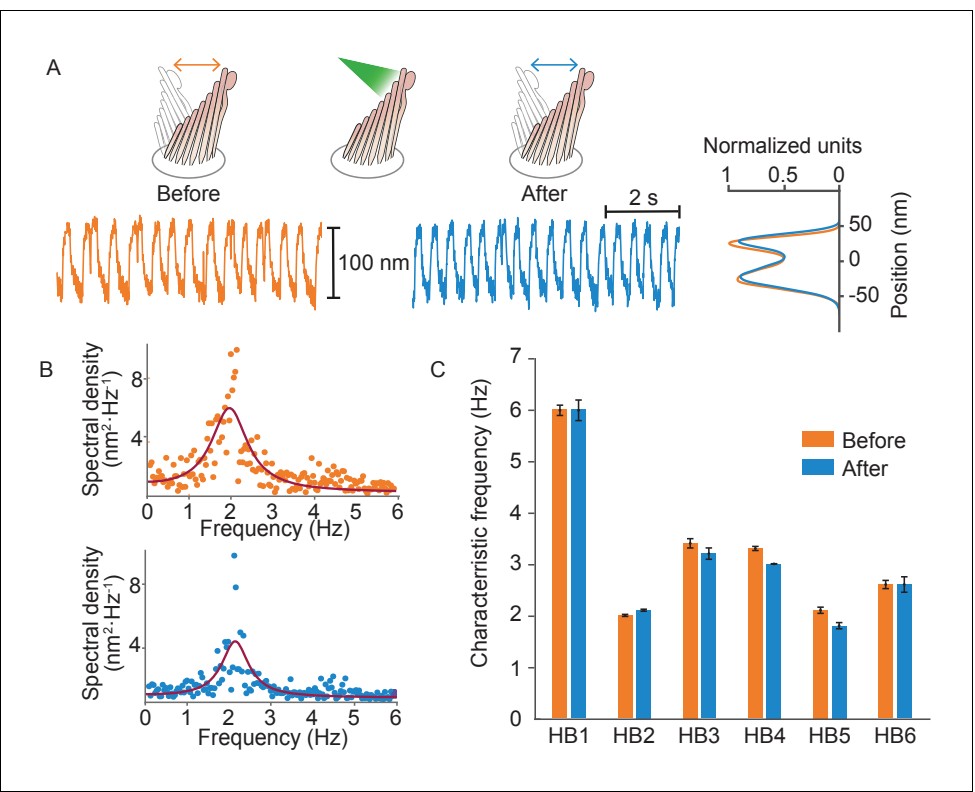

**Figure 5.** Normal operation of hair bundles after laser irradiation. (**A**) A hair bundle from the bullfrog displayed regular spontaneous oscillations prior to irradiation (orange). After 25 pulses of 30 mW, 561 nm light from a fiber 5 μm away, the bundle continued to oscillate with a similar waveform (blue). The histograms portray the distribution of bundle positions under the two conditions and confirm that the amplitude of oscillation was similar before and after irradiation. (**B**) The power spectrum of the same bundle's oscillations prior to irradiation shows a frequency peak around 2 Hz, as determined by a double-Lorentzian fit. The power spectrum after irradiation has a similar peak frequency. (**C**) Six hair bundles (HB1 - HB6) subjected to similar treatment showed insignificant changes in their peak frequencies of oscillation after irradiation. The bundle in panels (**A**) and (**B**) is HB2. The upper compartment of the experimental chamber contained endolymph in these experiments.

## Survival of mechanotransduction after laser irradiation

The hair bundles of healthy hair cells from the bullfrog can oscillate back-and-forth even in the absence of external stimulation (*Martin et al., 2003*). These spontaneous oscillations are a manifestation of the active process that these cells employ to amplify mechanical stimuli by counteracting viscous damping. The presence of spontaneous oscillations, which require a fully functional transduction apparatus, offers a means of assessing the viability of hair cells and the preservation of mechanotransduction following exposure to laser irradiation.

We compared the spontaneous oscillations of six hair cells before and after subjecting them to 25 pulses of light. Even at laser powers sufficient to deflect hair bundles over their entire physiological range of motion, the amplitudes and frequencies of the bundles' oscillation were unaffected by laser irradiation (*Figure 5*). This experiment demonstrates that the mechanotransduction apparatus was not damaged by our stimulation method.

To further assess the health of the hair bundles exposed to laser irradiation, we compared their uptake of FM1-43—a fluorescent dye that enters a hair cell through open mechanotransduction channels (*Gale et al., 2001*)—with that of the surrounding, unexposed hair cells and that of mechanically damaged bundles (*Appendix 1—figure 6*). The fluorescence signal from laser-irradiated hair bundles showed no visible difference with respect to that of the unscathed cells, whereas mechanically damaged bundles were visibly dimmer. This diminished fluorescence likely resulted from breakage of the tip links that reduced the opening of the mechanotransduction channels, thereby limiting the intake of the dye.

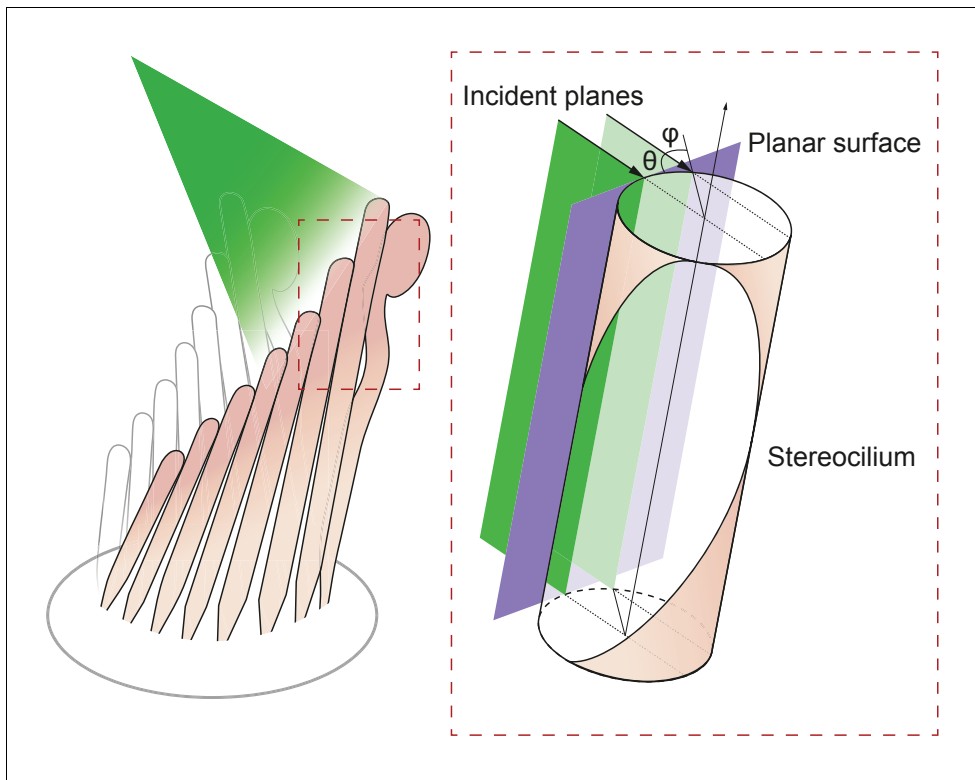

**Figure 6.** Geometry of irradiation of a cylindrical surface. As the beam from an optical fiber strikes a hair bundle, two representative sheets of the incoming light (green) are depicted along with the propagation direction of each (black arrows) and their angle of incidence θ with respect to the long axis of the stereocilium. Both incident planes are perpendicular to a plane tangent to the stereociliary surface (purple). Along its line of incidence onto the stereocilium, the centered sheet of light (dark green) is radially normal to the cylinder. The off-center sheet of light (pale green) strikes the stereocilium at an angle φ with respect to the normal.

## Discussion

We have used tapered optical fibers to apply both rapid and prolonged forces to individual hair bundles. The tip of each tapered fiber was small enough to be positioned near a hair bundle, and the ball lens at its tip restricted the divergence of the emitted light beam. Irradiation could be confined to a single bundle, whereupon the uniform illumination of all the stereocilia implied that each experienced a similar photonic force. It was also possible to irradiate only a portion of a bundle. Our control experiments indicated that even extensive irradiation of the magnitude necessary to evoke large displacements did not harm the hair bundles.

There are two difficulties with photonic stimulation. Although a routine procedure after the assembly of the necessary facilities, fabrication of a tapered optical fiber requires specialized equipment and a safe environment for the use of etching solution. A second issue is calibration: unlike the force delivered by a flexible fiber, which can be calibrated through the fiber's Brownian motion, the force exerted by photonic stimulation is not easily measured. The force can nonetheless be estimated by the use of targets whose stiffness has been independently determined, especially glass fibers such as those used in this study, or passive hair bundles including those subjected to chemical fixation.

The use of photonic stimulation offers at least five advantages over the currently used methods of mechanical stimulation. First, stimulation is rapid: in the present study, the rise time of mechanical responses was set by a hair bundle's stiffness and drag coefficient, without any effect of the drag on a stimulus fiber or the inertia of a piezoelectric actuator. Second, stimulation could be made still more rapid by a process analogous to 'supercharging' in a voltage-clamp system (*Armstrong and Chow, 1987*): transient irradiation with a very bright light could be used to deflect a bundle to a desired position, after which a steady force would be applied by weaker illumination during the measurement of a response. Because illumination can be switched off, a third virtue is that there is no possibility of an ill-defined steady-state offset in bundle position owing to mispositioning of a fiber or leakage from a fluid jet. The uniform illumination of the stereocilia in a bundle offers a fourth advantage, especially for mammalian hair bundles that exhibit relatively poor lateral coupling between stereocilia. And finally, photonic stimulation can be used in spaces too restricted to admit a flexible fiber or fluid jet. In particular, it should be possible to stimulate one or several hair bundles in preparations such as a hemicochlea (*He et al., 2004*) or an isolated cochlear segment (*Chan and Hudspeth, 2005a*; *Chan and Hudspeth, 2005b*).

## Materials and methods

### Estimation of photonic force

Each absorbed photon imparts all of its original momentum to the absorbing object and thereby provides an impulsive force. A reflected photon delivers twice the momentum provided by an absorbed one, whereas a refracted photon imparts momentum dependent on the angle of refraction. Because reflection sets the upper limit of the force that might be delivered to a hair bundle by a particular beam of light, we begin our analysis by treating the bundle as a perfect reflector. Averaged over one oscillation of the electromagnetic field, the radiation pressure due to illumination striking a hair bundle at an incident angle θ to the normal of the surface is *Paschotta, 2010*; *Hulst, 2003*:

$$P = 2\frac{<S>}{c}cos^2\theta \tag{1}$$

in which $P$ is the radiation pressure, $S$ is the average power of the electromagnetic wave, and $c$ is the speed of light in vacuum. *Equation 1* can alternatively be written in terms of irradiance $I$, or power per unit area, with units $\mathrm{W \cdot m^{-2}}$, and laser power (Pwr):

$$P = 2\frac{I}{c}cos^2\theta = 2\frac{Pwr}{A \cdot c}cos^2\theta \tag{2}$$

The force $F$ produced at an angle θ is then

$$F = 2\frac{Pwr}{c}cos\theta \tag{3}$$

For completely absorbed photons, this relation can be modified to

$$F = \frac{Pwr}{c}cos\theta \tag{4}$$

In a physiological solution, the refractive index is approximately 1.33, and therefore the speed of light is c/1.33. The angle of incidence in our experiments is 20°, which is set by the physical clearance between the objective lens and the preparation. By Snell's law, the angle of reflection is equal to the angle of incidence. Therefore, in the purely reflective case, using *Equation 3*, we estimate that 10 mW of laser power that impinges normal to the surface of the reflector generates approximately 80 pN of force. However, this upper limit of the force is not achievable because stereocilia are not perfect mirrors. The actual force experienced by a hair bundle depends on the difference of the refractive indices between the solution and the stereocilia, a larger difference indicating more reflected light and larger force. As discussed below, the angle of incidence is also important.

## Interaction of light with stereocilia: simple reflection and refraction

The interaction of light with stereocilia can be described by Fresnel equations that specify how the electric field vector's orientation, either parallel or perpendicular to the plane of incidence, determines the amplitude of reflection and transmission (*Figure 6*; *Born et al., 1999*).

$$R_\parallel = \frac{n_2 cos\theta_\mathrm{I} - n_1 cos\theta_\mathrm{T}}{n_2 cos\theta_\mathrm{I} + n_1 cos\theta_\mathrm{T}}A_\parallel$$

$$R_\perp = \frac{n_1 cos\theta_\mathrm{I} - n_2 cos\theta_\mathrm{T}}{n_1 cos\theta_\mathrm{I} + n_2 cos\theta_\mathrm{T}}A_\perp \tag{5}$$

$$T_\parallel = \frac{2n_1 cos\theta_\mathrm{I}}{n_2 cos\theta_\mathrm{I} + n_1 cos\theta_\mathrm{T}}A_\parallel$$

$$T_\perp = \frac{2n_1 cos\theta_\mathrm{I}}{n_1 cos\theta_\mathrm{I} + n_2 cos\theta_\mathrm{T}}A_\perp \tag{6}$$

In this set of equations, the transmission coefficient $T$ or reflection coefficient $R$ specifies the fraction of light either reflected or transmitted at the interface of two media. The subscripts $\parallel$ and $\perp$ denote the orientation of the electric filed, respectively parallel or perpendicular to the plane of incidence. Light of initial amplitude $A$ propagates from the medium of refractive index $\eta_1$ into that of refractive index $\eta_2$. The angles $\theta_\mathrm{I}$ and $\theta_\mathrm{T}$ are the angles of incidence and transmission (refraction), respectively. To estimate the refractive index of stereocilia we use the Gladstone-Dale relation (*Gladstone and Dale, 1863*)

$$n = n_0 + \alpha\rho \tag{7}$$

in which $\eta_0$ is the refractive index of the solution, $\alpha$ is the refractive index increment for protein (*Fasman, 2020*), 200 $m^3$•$kg^{-1}$, and ρ is the concentration of protein in a stereocilium, 250 kg•$m^{-3}$ (unpublished data). We expect the refractive index of the stereocilium to be approximately 1.4. The incident angle in our apparatus is 20°, so by use of Snell's law we find the angle of refraction for the transmitted light beam to be 19°. Applying these values to Fresnel's equations, we calculate the following coefficients:

$$R_\parallel = 0.022$$
$$R_\perp = 0.029$$
$$T_\parallel = 0.971$$
$$T_\perp = 0.971 \tag{8}$$

In view of the strong birefringence of stereocilia, we expect the photonic force to be greatest when the electric field is aligned parallel to a hair bundle's vertical axis. Taking into account only the

parallel components of the Fresnel equations, the coefficient of the reflected amplitude is $R_{\parallel} = 0.022$: approximately 0.05% of the power, or only 0.015 mW of the 30 mW incident on the stereocilia, should be reflected. The photonic force generated from reflection is therefore about 0.45 pN, a force unable to move a hair bundle appreciably. We must therefore reject a simple model of reflection and seek an understanding based on the reflective properties of curved surfaces.

## Interaction of light with stereocilia: reflection from a cylindrical surface

The reflectivity, or fraction of backscattered light, is significantly higher for a curved object than for a planar one (*Ashkin, 1970*). We may analyze this effect by considering the behavior of flat sheets of light incident upon a cylinder such as a stereocilium and parallel with its long axis (*Figure 7A*). Although a light sheet that strikes the stereocilium perpendicular to its surface exhibits only the effects discussed in the previous section, an off-center light sheet can produce a significantly greater force.

At any position along the stereocilium we may evaluate the behavior of representative rays of light as they impinge upon the front and back surfaces of the stereocilium. A ray exactly normal to the surface is partially reflected and partially transmitted, without refraction, through the stereocilium (*Figure 7B*). This ray exerts force on the stereocilium by reflecting from its front surface, with a lesser force provided by a fraction of the transmitted light that scatters from the back surface.

A ray that strikes the stereocilium at a modest distance from its center undergoes partial reflection at the front surface, thereby producing a force in the direction of propagation and toward the stereociliary axis (*Figure 7C*). Because the transmitted portion of the ray is incident upon the back surface of the stereocilium at an angle less than the critical angle for total internal reflection, it undergoes both reflection and refraction as it exits the stereocilium. That process again pushes the stereocilium in the direction of propagation as well as away from the midplane of the stereocilium.

A surprising effect ensues for a ray that impinges upon the stereocilium well away from its midplane. Such a ray undergoes partial reflection, pushing the stereocilium in the direction of propagation and toward its axis (*Figure 7D*). The refracted light then strikes the back surface of the stereocilium at an angle that exceeds the critical angle for total internal reflection, which—for stereociliary cytoplasm of refractive index $\eta_S \approx 1.4$ and water of $\eta_W = 1.33$—is approximately 72°. That ray exerts a force in the direction of light propagation and toward the stereociliary midplane. Moreover, before it eventually exits the stereocilium, the reflected ray might well undergo one or more additional total internal reflections, the first several of which exert additional force in the direction of propagation.

Because a stereocilium's diameter is similar to the wavelength of light, its optical properties cannot be described in detail by geometric optics, but involve calculations beyond the scope of this work. However, it has been shown that for a cylinder with an aspect ratio of 15, similar to that of a stereocilium of length 8 μm and diameter 0.5 μm, the reflectivity is about 3.5 times that of a sphere of equal volume (*Gordon, 2011*). Moreover, because stereocilia are closely spaced in a regular geometric array, they likely form a grating that exhibits complex interference effects. Nonetheless, even the qualitative description offered above emphasizes the importance of stereociliary curvature in providing exceptionally high reflection and unexpectedly great forces on stereocilia.

## Empirical estimation of photonic force

It is possible to estimate the photonic force that elicited a given hair-cell response from the stiffness of the moving hair bundle. This can be measured a posteriori—after the cell's response to light stimulation has been recorded—by displacing the same bundle mechanically with a flexible glass probe of known stiffness $k_p$, glued to the hair bundle's tip and driven by a piezoelectric actuator (*Benser et al., 1996*) (see *Appendix 1—figure 7*). The small pivoting angles of a hair-bundle responses allow us to treat the force delivered by the flexible fiber, $F_p = k_p(\Delta_{\text{probe}} - \Delta_{\text{tip}})$, and the reaction of the hair bundle, $F_{\text{HB}} = k_{\text{HB}}\Delta_{\text{tip}}$, as hookean elastic forces. Here $\Delta_{\text{probe}}$ is the displacement imposed on the flexible fiber's base by the actuator, and $\Delta_{\text{tip}}$ is the movement of its tip. The latter is identical to the motion of the hair bundle to which it is attached, and can be tracked with a photodiode.

We used this approach to estimate the force delivered by photonic pressure onto a hair bundle of the bullfrog's sacculus. A 50 ms pulse of photonic pressure at 12.5 mW output power elicited a

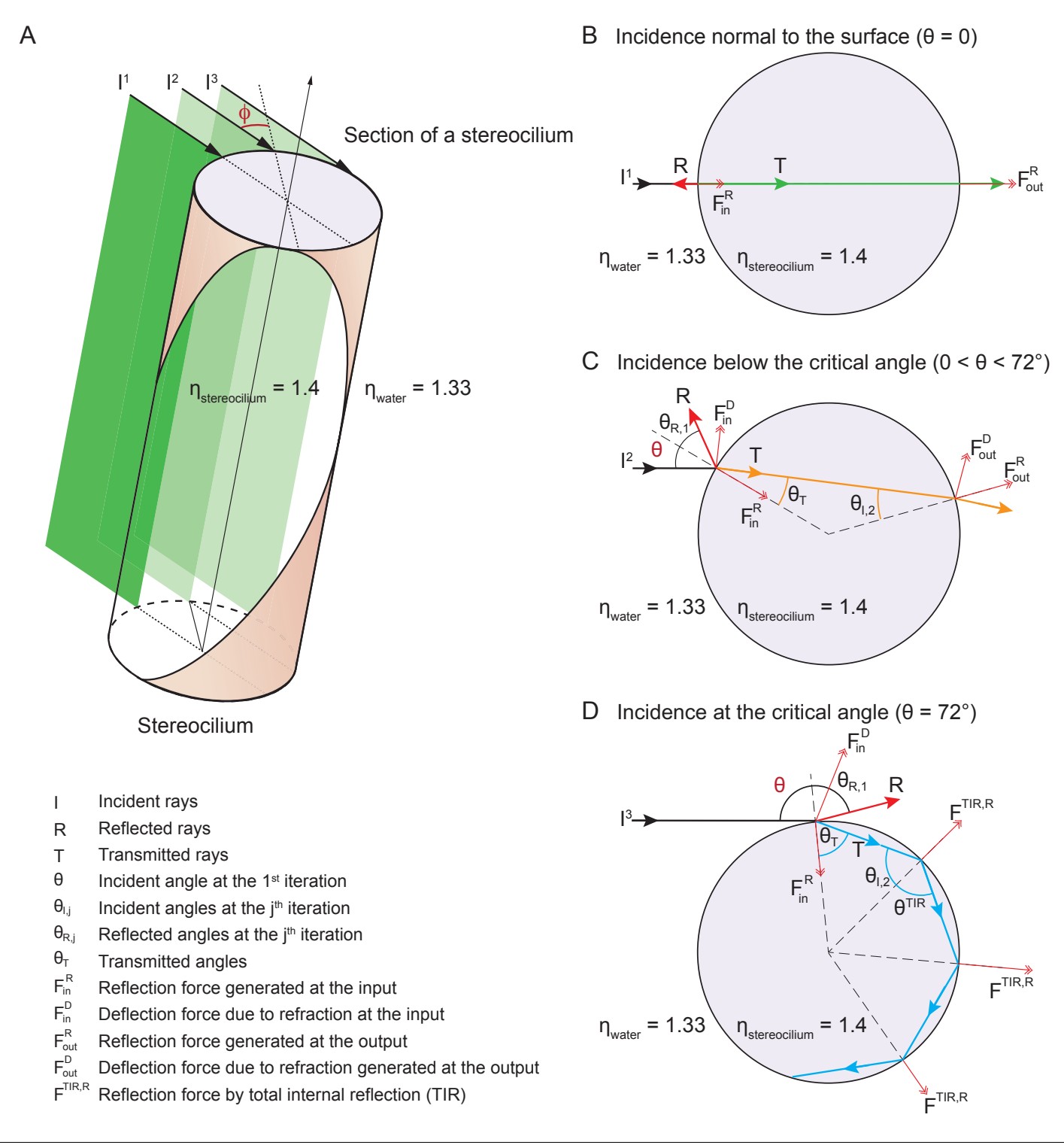

**Figure 7.** Potential fates of a plane wave incident on a stereocilium. (**A**) Three representative rays of a light beam interact with a stereocilium; I, R and T denote the incident, reflected, and transmitted portions of each ray. All superscripts and subscripts are defined in the figure. The rays $I^1$, $I^2$, $I^3$ (black arrows) indicate the direction of light in water (refractive index 1.33) as it strikes a stereocilium whose refractive index is 1.4 and whose section is shown in lavender. The ray $I^1$ is incident along the normal to the stereocilium, the axis of symmetry of the section. For parallel rays further from $I^1$, the angle of incidence $\phi$ at which the light strikes the stereocilium's surface increases as measured with respect to the normal. These three rays of incident light impart distinct forces on the stereocilium. (**B**) When a ray is reflected, it forces the stereocilium in the opposite direction and the direction of this input reflection force $F_{in}^R$ is radially aligned with the center. (**C**) If the ray is deflected due to refraction, a deflection force ($F_{in}^D$) is generated on the stereocilium
*Figure 7 continued on next page*

*Figure 7 continued*

that is perpendicular to the direction of the ray as it propagates within the stereocilium. The incident angle is equal to the reflection angle, as is the case for the ray $I^2$ as it first strikes the stereocilium ($\theta = \theta_{R,1}$). The light that is refracted propagates along T (orange line) once inside the stereocilium until it reaches the boundary with water. At this second collision the incident angle $\theta_{I,2}$ is equal to the refractive angle $\theta_T$, which is too small to cause another reflection; as a result, the ray exits into water and no deflection force is generated. (D) A third kind of force arises if total internal reflection (TIR) occurs, as happens when the angle of the incident light beam is such that a ray remains trapped inside the stereocilium as it is repeatedly reflected at the boundary with water. In the case of ray $I^3$, the incident angle is equal to the critical angle for total internal reflection—72° in this case—and the light remains within the stereocilium as T (blue arrow) and is reflected repeatedly each time it reaches the boundary with water. Three successive total internal reflections are shown; each generates a reflection force $F^{TIR,R}$.

20 nm deflection of the hair bundle, whose stiffness was subsequently measured to be $1.8 \pm 0.2$ mN•m$^{-1}$ (mean ± SEM)—a relatively stiff hair bundle. This allowed us to estimate that approximately 40 pN was delivered to the hair bundle by radiation pressure.

## Fabrication of a tapered optical fiber

In order to produce an optical fiber with a tip small enough to approach an individual hair bundle, it is necessary to thin the fiber's 60 µm-thick cladding to expose the inner core of 5 µm diameter. Various methods have been employed to reduce the diameter of fibers in near-field optical microscopy and in the development of optical-fiber sensors. One common method is to use a carbon dioxide laser to machine optical fibers (*Ozcan et al., 2007*). Although this method is capable of creating symmetrical fibers, $CO_2$ lasers are expensive and require complex optics. Two other methods used for removing material in optical fibers are femtosecond laser micromachining (*Wei et al., 2008a*; *Wei et al., 2008b*; *Liao et al., 2012*; *Yuan et al., 2012*) and focused-ion-beam milling (*Kou et al., 2010*; *Yuan et al., 2011*; *André et al., 2014*). Although both methods are effective, they are time-consuming and require expensive instruments.

On the basis of previous experiments with glass fibers, we suspected that the interaction of tapered fibers with living specimens would contaminate the fibers' tips and thus limit the use of each fiber to only a few experiments. Furthermore, the gradual degradation due to several hundred high-power optical pulses during an experiment would limit a fiber's use to a few experiments. Both considerations required that fibers be tapered easily and cost-effectively in a typical laboratory setting. We created tapered optical fibers by Turner's wet chemical etching with hydrofluoric acid (*André et al., 2014*). With this method, a fiber can be shaped in about 1.5 hr in any laboratory with a fume hood and few tools. In shaping each fiber, we started with a single-mode optical fiber 1 m in length and with an FC/PC connector at one end. The distal end of the fiber was prepared by stripping a 12 mm length of its polymeric jacket and the polyamide coating and cleaning it with 70% ethanol. We inserted the fiber's end into a holder that allowed it to be attached to manipulators during the fabrication process.

Etching was conducted in a fume hood (*Figure 8*). After 47.5 mL of concentrated (48%) hydrofluoric acid had been placed in a polypropylene tube (Corning Life Sciences, Tewksbury, MA, USA), 2.5 mL of red kerosene oil was added. The oil layer's purpose was twofold. First, it provided protection to the fiber above the surface from attack by acid vapor. Second, the height of the aqueous meniscus was dependent upon the diameter of the immersed fiber, and thus declined as etching proceeded. When etching was complete, the oil layer isolated the tip from the acid.

The fiber's holder was attached to a motorized linear actuator (Nanotec Electronic GmbH and Co KG, Feldkirchen, Germany) with 3 µm positioning resolution and the height of its tip was controlled through a computer interface (LabVIEW; National Instruments, Austin, TX). Because the diameter of the fiber's tip at any point along its length depended on the duration of its immersion, it was critical to control the fiber's extraction speed. For maximal stability during experiments, we set the length of the taper to 8 mm, the minimum required for reliable clearance of the objective lens.

After coupling a 633 nm wavelength laser to the optical fiber to render its tip visible during etching, we lowered the fiber until its tip was immediately above the interface between the oil and the acid. Under computer control, the actuator then performed a series of insertions and extractions of the fiber. The initial program inserted the fiber 10 mm into the acid at 2 mm•s$^{-1}$ and extracted 8mm at the same speed. The routine next extracted the optical fiber for 20 min at 37.5 µm•min$^{-1}$, reducing its diameter from 125 µm to 60 µm. The extraction then stopped and the fiber remained in the

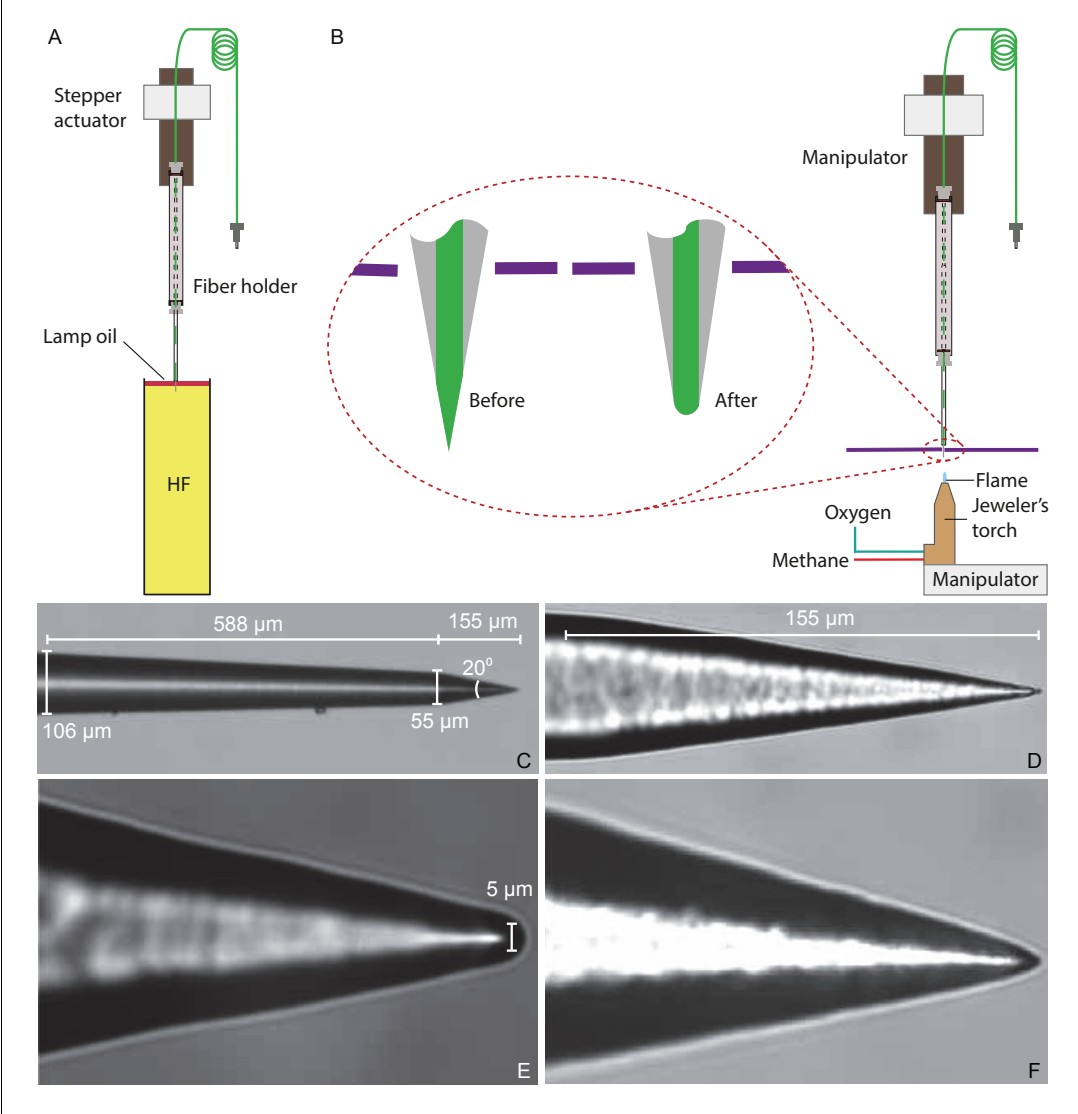

**Figure 8.** Preparation of a tapered optical fiber. (**A**) A schematic diagram depicts the process of reducing the optical fiber's diameter by chemical etching. The tapered optical fiber is attached to the shaft of a linear actuator and lowered into a tube of hydrofluoric acid (HF) topped with a thin layer of lamp oil (kerosene). The convergence angle of the fiber's tip is determined by the rate at which the tip is extracted from the acid. (**B**) A schematic diagram depicts the apparatus for creating a hemispherical lens at the fiber's tip. With the aid of a three-axis micromanipulator, the fiber's tip is inserted through a hole less than 1 mm in diameter in a horizontally mounted metal plate (purple line). The nozzle of a jeweler's torch is aligned with the optical fiber by means of a second micromanipulator. Careful adjustment of the flow of oxygen and methane yields a flame about 0.5 mm in height. Under microscopic observation, the flame is raised until the fiber's tip melts, whereupon the fiber is immediately retracted. (**C**) An image of a fiber's tip after chemical etching with 48% hydrofluoric acid and before polishing shows a slow taper over 588 µm followed by a steep taper over the final 155 µm. (**D**) At a higher magnification, the tapered but unpolished tip displays a cone angle of about 20°. (**E**) A high-magnification image depicts a polished tip with a relatively large hemispherical lens. (**F**) Another polished tip ends in a narrower lens.

acid for 18 min, during which tip was etched at a steeper angle by the gradual fall of the meniscus. The fiber was rinsed with distilled water and then with isopropyl alcohol and air-dried in the fume hood.

## Creation of a miniature hemispherical lens

When light exits an optical fiber into a medium of lower refractive index, such as water, it diverges rapidly (*Kohls and Holst, 1998*). To minimize this divergence and direct the light to fall evenly upon a hair bundle, we created a focusing lens at the fiber's tip. Although it is a common practice to

attach microscopic lenses to optical fibers with flat, polished ends (*Liberale et al., 2010*; *Eversberg and Vollmann, 2015*), it was not practical do so with a taped optical fiber ending in a sharp point. We therefore created a lens by melting the fiber's tip of silicon dioxide, which melts at 1713 °C (*Haynes, 2011*).

We used a jeweler's torch with a nozzle 250 µm in diameter and fed with pressurized methane and oxygen. Creating a lens required clear visualization of the fiber's sharp tip and precise manipulation of the torch (*Figure 8B*). Using a pair of manual manipulators, we mounted the torch below the fiber's tip and visualized them with a horizontal microscope. Because the hot air rising from the torch caused the thin tip of the fiber to flutter, we reduced the convection around the fiber by partially exposing the tip to the torch's flame through an aperture 1 mm in diameter in a square metal plate 50 mm on each side. After focusing the image of the fiber's tip on an eyepiece reticle, we carefully raised the unlit torch toward the fiber and aligned the two to prevent asymmetry in the lens. The torch was then lowered, lit, and adjusted to a flame height of about 0.5 mm. As we then raised the torch, the core of the fiber began to melt and promptly assumed a hemispherical shape (*Figure 8C–F*). We immediately lowered the torch and allowed the fiber to cool before removing it from the apparatus.

## Estimation of the area of irradiation

After fabricating a tapered optical fiber of suitable shape, we characterized its pattern of illumination before using it in experiments. This process was designed to evaluate the optimal distance between the fibers tip and a hair bundle so that we could match the diameter of the light spot to the bundle's width.

After coupling a laser to the tapered fiber, we approximated the fiber's tip to the flat end of an ordered fiber bundle and monitored their separation under a microscope with a camera (*Figure 9A*). Passing through a droplet of water, the light from the tapered fiber impinged on the fiber cores of the bundle and propagated to the distal end, where it was imaged through a dry objective lens (Plan 10X, numerical aperture 0.25, Olympus, Tokyo, Japan) onto second camera. The illuminated

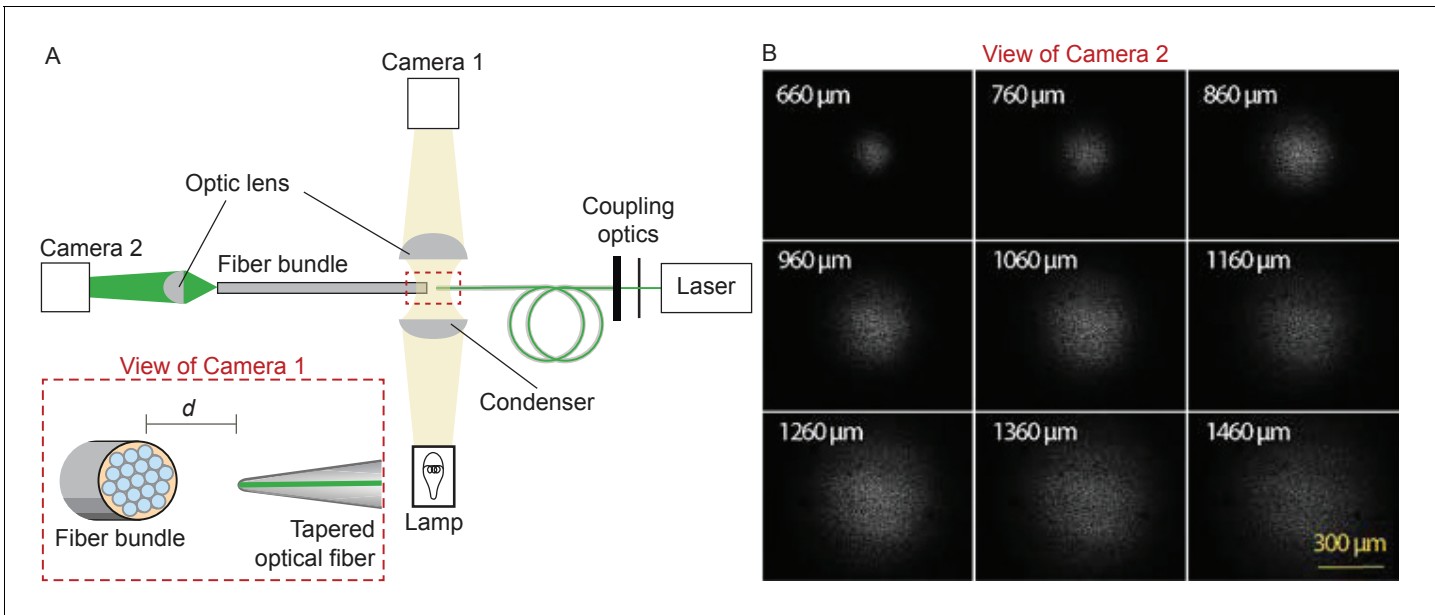

**Figure 9.** Characterization of the light spot produced by a polished fiber. (**A**) A schematic diagram depicts the apparatus for characterization of a fiber's output. During observation by camera one attached to the microscope, the tapered optical fiber is coupled to the laser and brought near the transverse surface of a fiber bundle (IGN-8/30, Sumitomo Electric, Japan). The view through camera 1, with the fiber tip pointing at one end of the fiber bundle a distance d away, is schematized in the inset. The fiber bundle has 30,000 inner cores, each 2 µm in diameter and with a center-to-center spacing of 4 µm. The distal end of the bundle is imaged by camera two at a magnification of 11.6X. (**B**) Images of the output, as captured with camera 2, show the divergence of the light beam as the tapered optical fiber is brought approximately 660 µm from the bundle and retracted by intervals of 100 µm.

fiber cores defined the diameter of the illuminated area on the fiber bundle (*Figure 9B*). By capturing images at intervals of 100 μm as the tip of the tapered fiber was withdrawn from the fiber bundle, and measuring the diameter of the illuminated area at 95% of the power spread, we estimated the divergence angle of the light cone.

## Experimental configuration

Animal experiments were conducted with the authorization and according to the protocols of the relevant institutions. Except where specified, sacculi from adult American bullfrog (*Rana catesbeiana*) of both sexes were dissected and maintained in artificial perilymph (*Alonso et al., 2020*). Cochleas dissected from Sprague-Dawley rats four to twelve days of age and of both sexes were maintained in a saline solution comprising 127.5 mM NaCl, 10 mM sodium pyruvate, 1.2 mM $MgCl_2$, 5 mM D-glucose, 3 mM KCl, 0.5 mM $CaCl_2$, 1 mM creatine, 20 mM sucrose, 0.5 mM $NaH_2PO_4$, and 2 mM $Na_2SO_4$.

During each experiment, the tapered optical fiber was inserted through a glass capillary placed in a custom-made electrode holder that could be affixed to a micromanipulator (*Figure 10A*). This holder ensured that the fiber's tip was stable despite possible vibrations or displacements of the remainder of the fiber.

Under the control of a micromanipulator (ROE 200, Sutter Instruments, Novato, CA, USA), the fiber's distal end was introduced into the experimental chamber beneath a 60X water-immersion objective lens (LUMPlanFL N, numerical aperture 1.0, Olympus, Tokyo, Japan). The incidence angle of about 20° with respect to the horizontal ensured that the fiber cleared both the upper edge of the experimental chamber and the lower rim of the lens (*Figure 10B*).

Light from a 600 nm light-emitting diode (Prizmatix Ltd., Southfield, MI, USA) illuminated the specimen through an inverted 60X water-immersion objective lens (LUMPlan Fl/IR, numerical aperture 0.9, Olympus) that served as a condenser. To permit differential-interference imaging, a

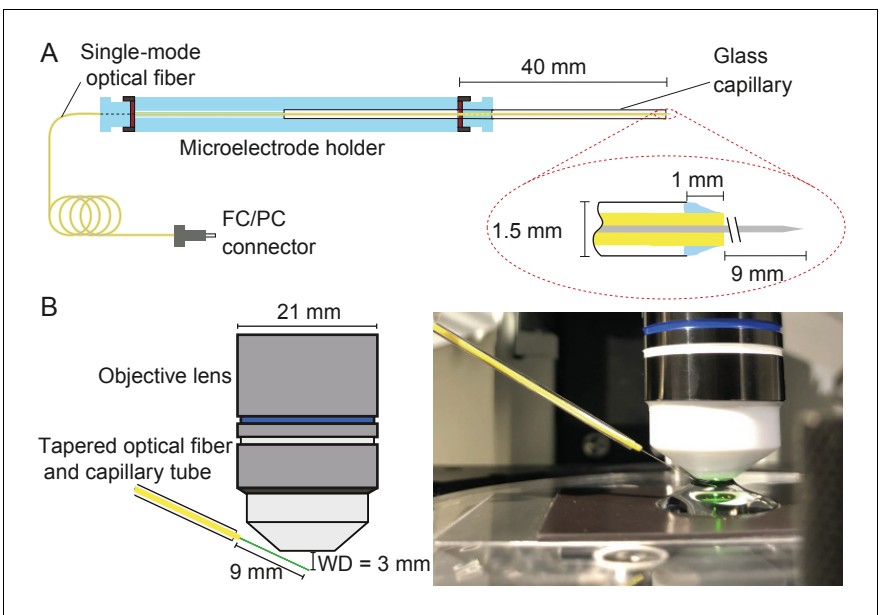

**Figure 10.** Positioning fiber under an objective lens using a custom-made holder. (**A**) Holder for the tapered optical fiber. The schematic drawing portrays a tapered optical fiber inserted in the mount constructed from a microelectrode holder and a glass capillary, from which the fiber's distal tip protrudes 10 mm. The coiled optical fiber's inner core is depicted in gray inside the yellow jacket. As seen in the inset, the space between the glass capillary and the yellow jacket that protrudes 1 mm past the capillary's tip is packed with vacuum grease (light blue). The distal end of the fiber is terminated with an FC/PC connector. Positioning of a fiber under an objective lens. (**B**) A schematic drawing (left) shows the length of the glass capillary that protrudes from the fiber-holder relative to the objective lens of working distance (WD) 3 mm. The photograph (right) shows the tapered optical fiber, objective lens of the microscope, and preparation chamber in an experiment.

polarizer was positioned just above the microscope's field diaphragm and a crossed analyzer above its tube lens, and both objective lenses were equipped with Wollaston prisms. A rotating quarter-wave plate above the polarizer permitted optimization of the image, and a heat filter protected the specimen from infrared damage. Light that had traversed the specimen, the objective lens, and the tube lens was relayed by two mirrors and projected with a total magnification of 900X onto a dual photodiode, which permitted measurements of hair-bundle movement with nanometer precision. A dichroic mirror imposed before the photodiode prevented contamination of the movement signal by light from the stimulating laser. For the selection of appropriate hair bundles, the light path could be diverted to a camera that permitted observation of the specimen on a digital monitor.

## Acknowledgements

We thank Brian Fabella for assistance with the fabrication of apparatus and the programming of data-acquisition software and Rodrigo Alonso for the preparation of bullfrog sacculi. The members of our research groups graciously provided critical comments on the manuscript. AJH is an Investigator of Howard Hughes Medical Institute. ASK was supported by grants 108034/Z/15/Z and 214234/Z/18/Z from the Wellcome Trust and by an Imperial College Network of Excellence Award.

## Additional information

### Funding

| Funder | Grant reference number | Author |
| --- | --- | --- |
| Howard Hughes Medical Institute | | AJ Hudspeth |
| Wellcome Trust | 108034/Z/15/Z | Andrei S Kozlov |
| Wellcome Trust | 214234/Z/18/Z | Andrei S Kozlov |

The funders had no role in study design, data collection and interpretation, or the decision to submit the work for publication.

### Author contributions

Sanjeewa Abeytunge, Data curation, Formal analysis, Validation, Investigation, Visualization, Methodology, Writing - original draft; Francesco Gianoli, Data curation, Formal analysis, Investigation, Visualization, Writing - review and editing; AJ Hudspeth, Conceptualization, Data curation, Supervision, Funding acquisition, Validation, Investigation, Methodology, Project administration, Writing - review and editing; Andrei S Kozlov, Conceptualization, Data curation, Supervision, Funding acquisition, Investigation, Writing - original draft, Project administration, Writing - review and editing

### Author ORCIDs

Francesco Gianoli https://orcid.org/0000-0002-4876-7978
AJ Hudspeth https://orcid.org/0000-0002-0295-1323
Andrei S Kozlov https://orcid.org/0000-0003-1993-8341

### Ethics

Animal experimentation: All procedures were conducted according to the rules and regulations of the home institution. At The Rockefeller University the procedures were approved by the Institutional Animal Care and Use Committee. At Imperial College London, the procedures were carried out in accordance with the U.K. Home Office Animals (Scientific Procedures) Act (1986).

### Decision letter and Author response

Decision letter https://doi.org/10.7554/eLife.65930.sa1
Author response https://doi.org/10.7554/eLife.65930.sa2

## Additional files

### Supplementary files

• Transparent reporting form

### Data availability

All source data can be found on Dryad: Citation Abeytunge, Sanjeewa; Gianoli, Francesco; Hudspeth, AJ; Kozlov, Andrei S. (2021), Rapid mechanical stimulation of inner-ear hair cells by photonic pressure, Dryad, Dataset, https://doi.org/10.5061/dryad.76hdr7sww.

The following dataset was generated:

| Author(s) | Year | Dataset title | Dataset URL | Database and Identifier |
|---|---|---|---|---|
| Abeytunge S, Gianoli F, Hudspeth AJ, Kozlov AS | 2021 | Rapid mechanical stimulation of inner-ear hair cells by photonic pressure | https://doi.org/10.5061/dryad.76hdr7sww | Dryad Digital Repository, 10.5061/dryad.76hdr7sww |

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

## Appendix 1

### Deflection of glass rods by photonic force

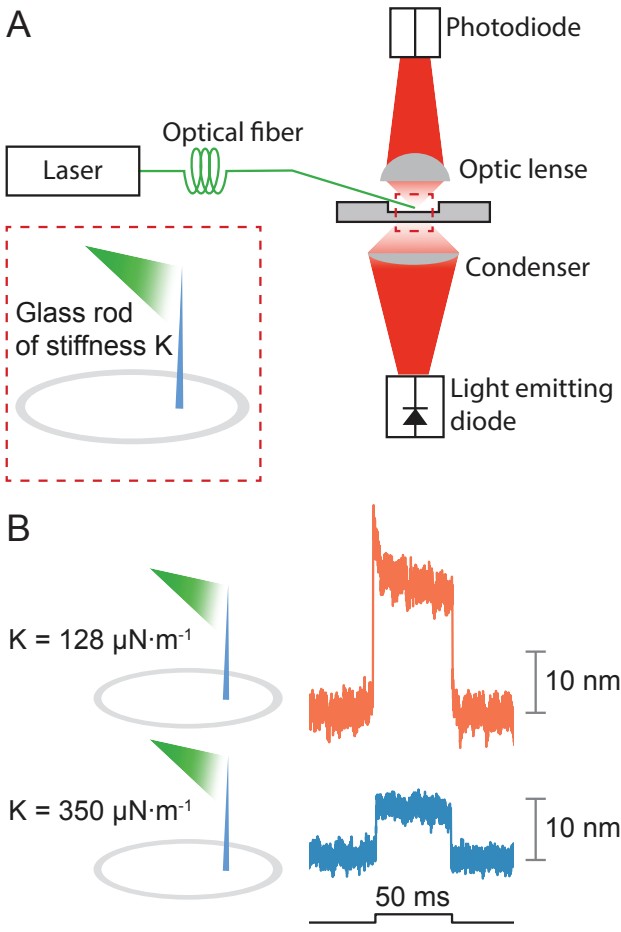

**Appendix 1—figure 1.** Application of photonic force applied to glass rods. (**A**) Two glass rods were placed in the experimental chamber and irradiated through a tapered optical fiber for 50 ms. The average of 25 deflections was recorded for each rod. (**B**) The glass rod with lower stiffness of 128 µN•m$^{-1}$ (orange) moved thrice as far as the fiber with a higher stiffness of 350 µN•m$^{-1}$ (blue). The estimated power of irradiation falling upon each rod was 20 mW at a wavelength of 561 nm. The sudden movement at the onset of illumination for the rod of lower stiffness likely stemmed from thermoelastic effects.

## Experimental configuration

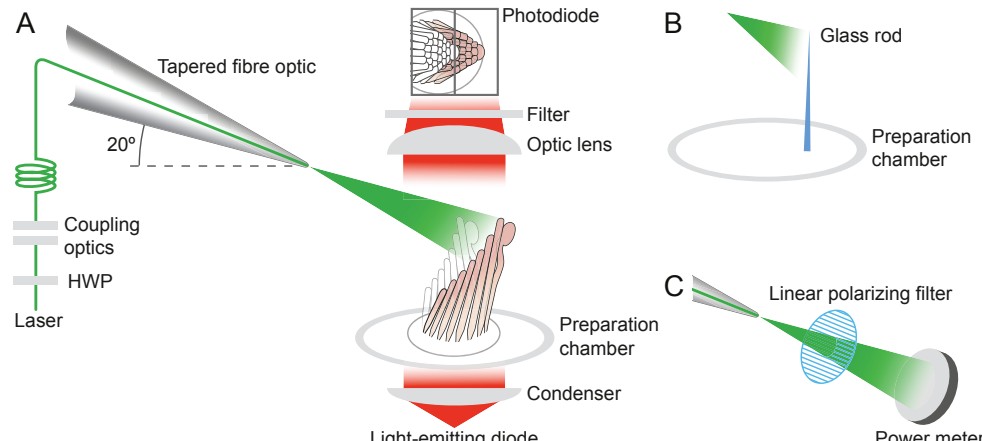

**Appendix 1—figure 2.** Experimental configuration. The schematic drawing shows the arrangement of the main components in the experimental apparatus. (**A**) The laser beam of 561 nm wavelength (green line) traverses the half-wave plate (HWP) and is coupled to the tapered optical fiber. The fiber's distal end with the microlens is approximated to the experimental preparation under a microscope. Illumination from a light-emitting diode with a central wavelength of 660 nm (red) is focused by a condenser onto a hair bundle, which is imaged onto a dual photodiode with a 60X objective lens of numerical aperture of 1.0. A dichroic mirror blocks 562 nm laser light but it passes the light coming from the light-emitting diode. (**B**) In control experiments, a glass rod (blue triangle) of stiffness comparable to that of a bullfrog's hair bundle is mounted vertically in the experimental chamber. The laser light (green triangle) deflects the rod, whose motion can be measured with the system in panel A. (**C**) To orient the plane of polarization of light with the long axis of the stereocilia, light from the tapered optical fiber is passed through a linear polarizing filter and its intensity is measured with a power meter. The maximal power is detected when the polarization plane is aligned with the transmission direction of the polarizing filter.

## Polarization dependence of hair-bundle responses

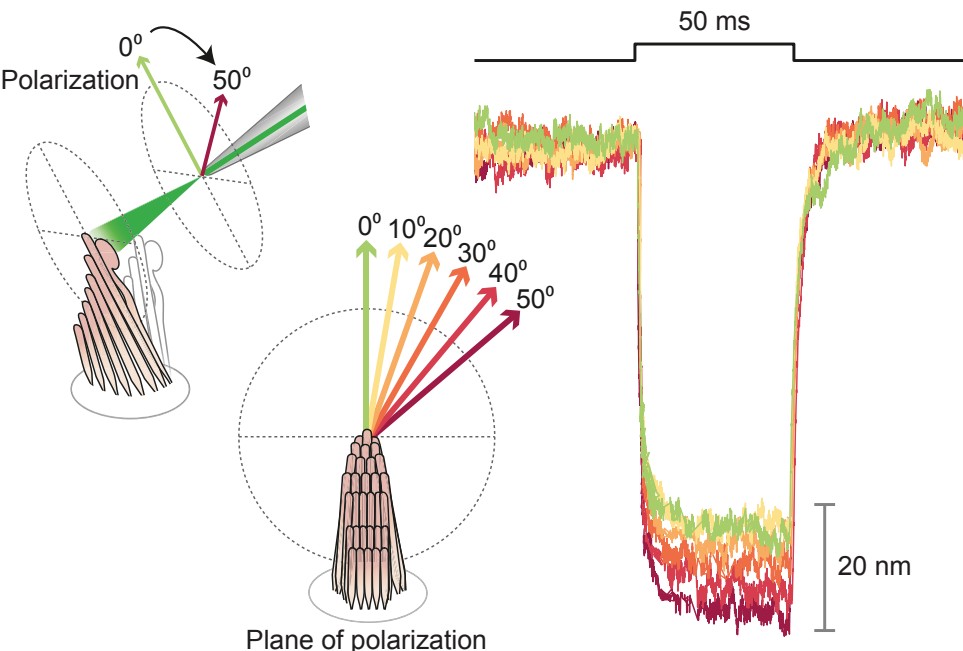

**Appendix 1—figure 3.** Effect of polarization on response amplitude. After its tip links had been broken by exposure to 5 mM BAPTA for $30\,\mathrm{s}$, a hair bundle from the bullfrog's sacculus was stimulated in the negative direction with 50 ms, 30 mW laser pulses. Using a half-wave plate between the laser and the coupling optics, we rotated the polarization plane about the axis of propagation between 0 and 50. For a simple polarized object, the reflected power should decline by the cosine of the angle. Although the results showed a qualitative agreement with the prediction, we observed a significantly smaller reduction in amplitude consistent with the fact that stereocilia are birefringent, but exhibit significant scattering of light at all angles. Each trace represents the average of 25 recordings.

**Stimulus diversity: steps and ramps**

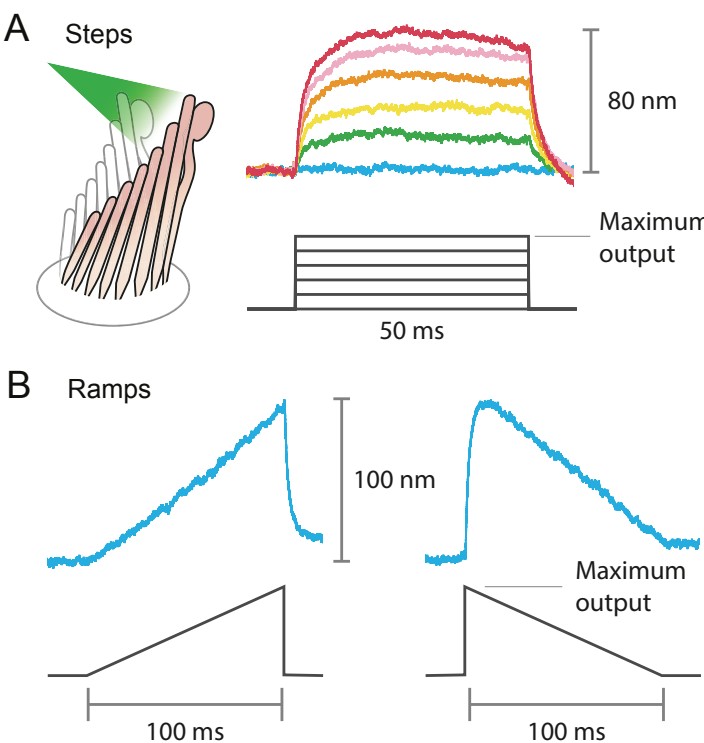

**Appendix 1—figure 4.** Example of the variety of stimuli offered by photonic-force stimulation. (**A**) Frog hair bundles were stimulated in the positive direction with 561 nm light. The fiber's power output was increased in five steps from 0 mW to the maximum, 12.5 mW for this fiber. Each colored trace shown is the average of 25 responses tracking the hair-bundle movement to a pulse of 50 ms at a constant laser power. (**B**) Frog's hair-bundle responses to increasing (left) and decreasing (right) ramps of 100 ms, in which the laser power was varied continuously between 0 mW and 12.5 mW.

### Separating the photothermal movement

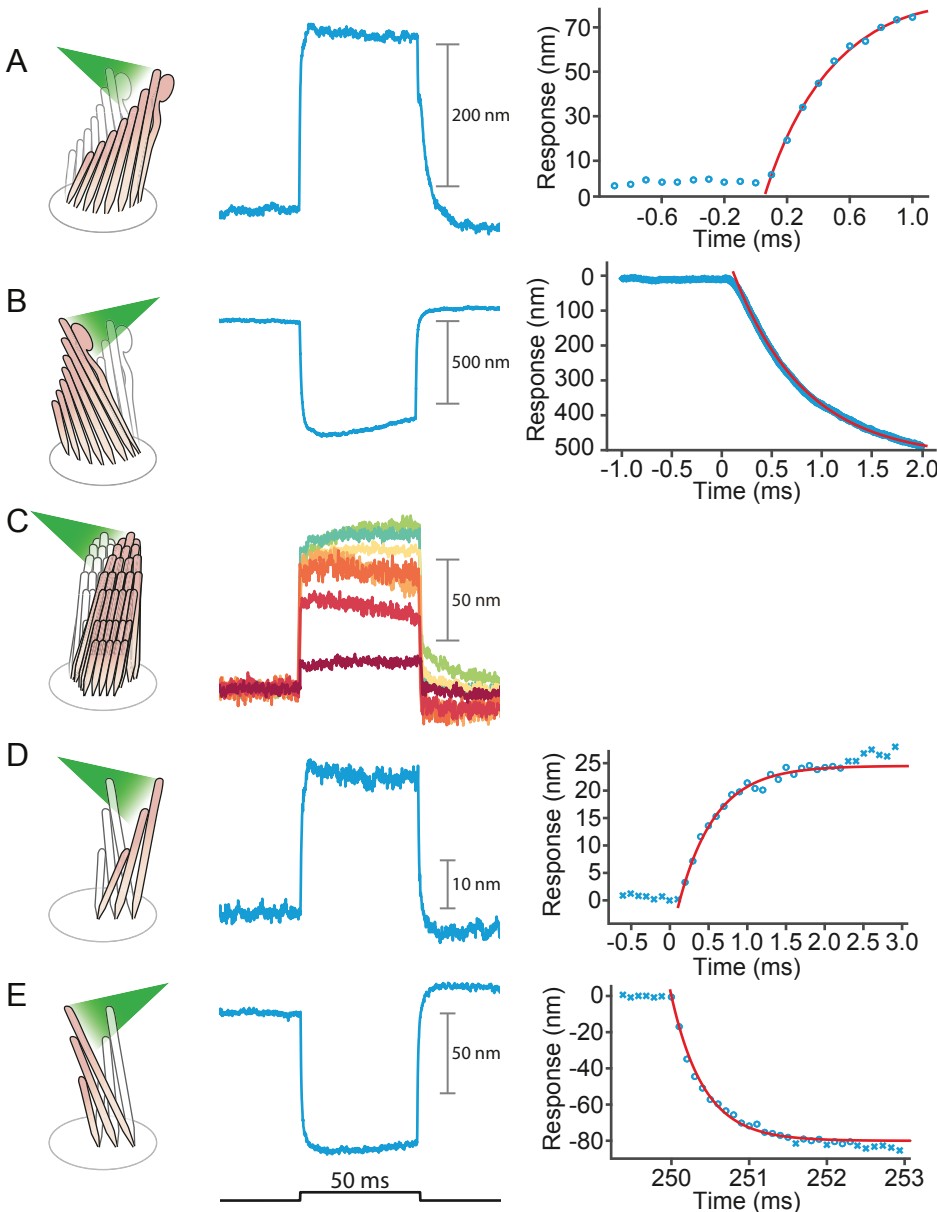

**Appendix 1—figure 5.** Deflection of hair bundles by optical radiation force without a photothermal effect. (**A**) After tip links had been ruptured by a Ca²⁺ chelator (BAPTA), photonic force displaced a bullfrog's bundle in the positive direction with a time constant of 415 s. In this and the other panels, the bundles were stimulated at 561 nm with 30 mW of input power and the records represent the average of 25 repetitions. (**B**) Stimulation in the negative direction evoked a negative movement with a time constant of 750 s. (**C**) Photonic force applied at 90° to the axis of sensitivity displaced a hair bundle in the direction of irradiation. (**D**) After the disruption of tip links, the hair bundle from a rat's outer hair cell moved with a time constant of 467 s in the direction of photonic stimulation. (**E**) Negatively directed stimulation conversely evoked motion in the negative direction with a time constant of 418 s. The upper compartment of the experimental chamber contained endolymph in these experiments.

## Survival of mechanotransduction after laser irradiation

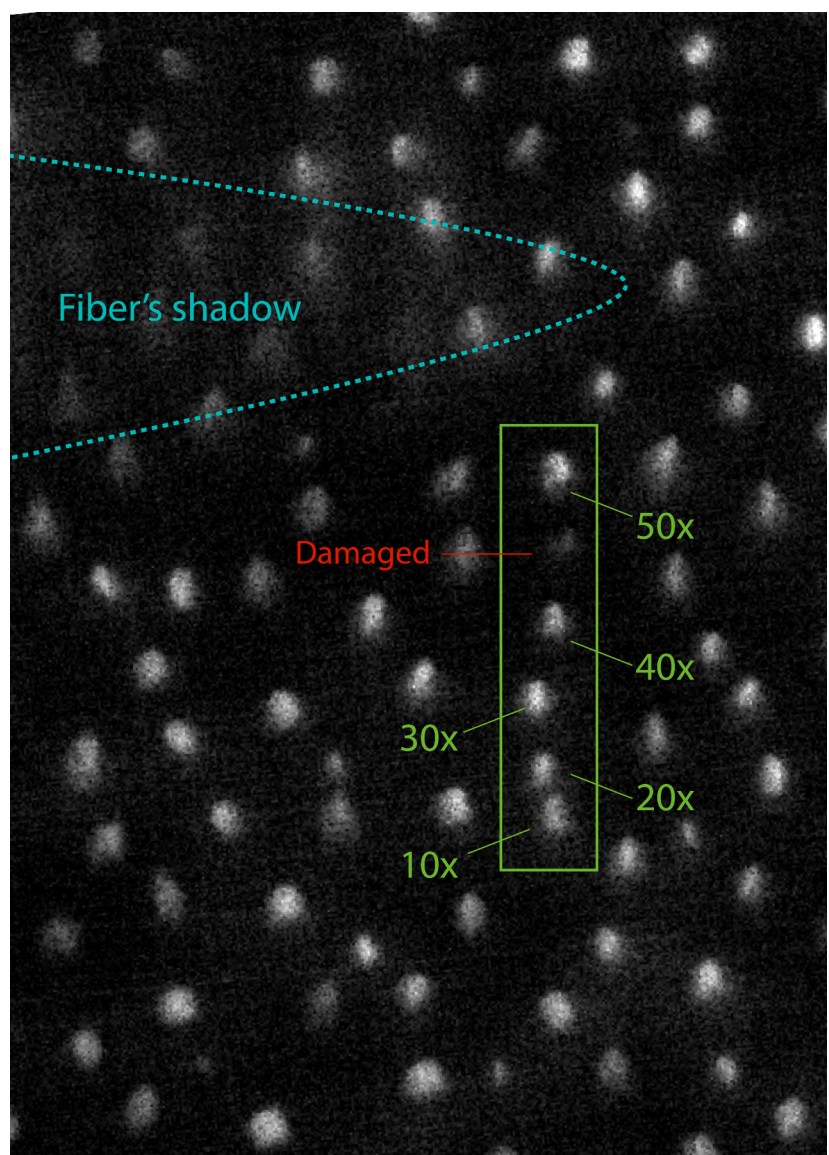

**Appendix 1—figure 6.** The fluorescence signal of bullfrog saccular hair bundles loaded with FM1-43 after exposure to laser irradiation. We stimulated successive hair bundles in a row with 50 ms pulses at a power of 12.5 mW, the full power available for the fiber. The five hair bundles in the green box were subjected to 10, 20, 30, 40, and 50 pulses, as indicated. A sixth hair bundle in the same row, labeled in red, was damaged by being crushed a few times by the end of the optical fiber. Immediately after those procedures, the sample was exposed to 1 mM of FM1-43 for one minute, after which fluorescence was recorded. The stimulated hair bundles show comparable levels of brightness between each other and with respect to those in the surroundings. The loading of the dye is visibly reduced in the mechanically damaged cell owing to breakage of the tip links.

## Estimation of the photonic force acting on a hair bundle

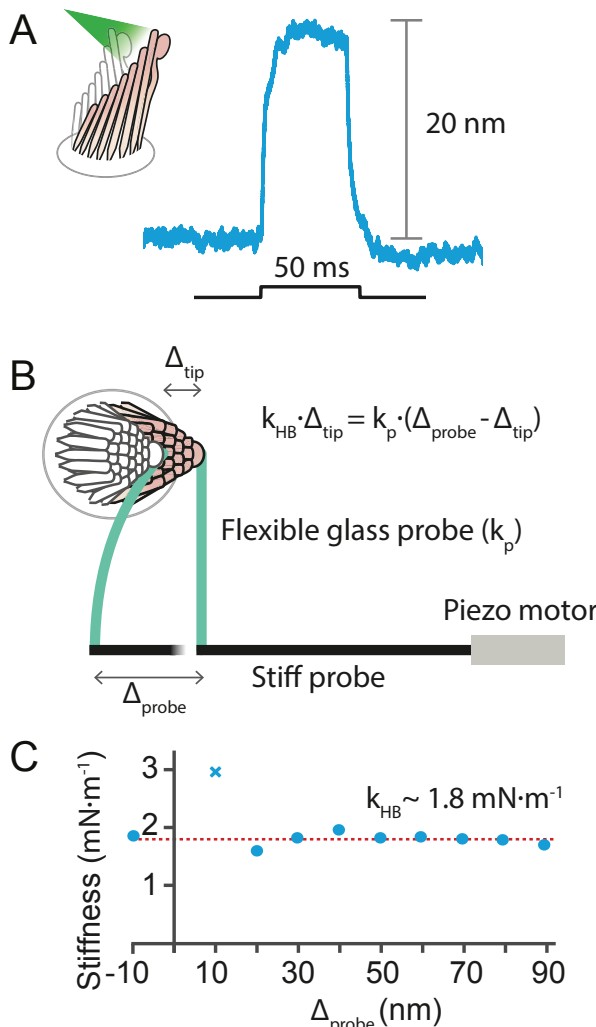

**Appendix 1—figure 7.** Estimation of the force exerted by radiation pressure onto a hair bundle of known stiffness. (**A**) When a hair bundle from the frog's sacculus was deflected by a 50 ms laser pulse at 12.5 mW output power, the bundle moved positively by 20 nm. The trace shown is the average of 25 repetitions. (**B**) The kinocilium of the same hair bundle was connected to the tip of a flexible glass probe of known stiffness $k_{\mathrm{p}}$, whose other end connected to a stiff probe driven by a piezoelectric actuator. To ensure adhesion between the glass and the kinocilium, concanavalin A was used to coat the flexible fiber. The motion of the fiber's tip $\Delta_{\mathrm{tip}}$ and the hair bundle was tracked with a dual photodiode. A movement of the stiff probe by $\Delta_{\mathrm{probe}}$ translates into a force $F_{\mathrm{p}} = k_{\mathrm{p}}(\Delta_{\mathrm{probe}} - \Delta_{\mathrm{tip}})$ delivered by the flexible probe onto the hair bundle, whose reaction force depends on its stiffness $k_{\mathrm{HB}}$ times the bundle's displacement $\Delta_{\mathrm{tip}}$. The balance of forces provides a way to estimate the hair-bundle stiffness: $k_{\mathrm{HB}} = k_{\mathrm{p}}(\Delta_{\mathrm{probe}}/\Delta_{\mathrm{tip}} - 1)$. (**C**) The stiffness of the hair bundle stimulated was measured by varying $\Delta_{\mathrm{probe}}$ between −10 nm and 90 nm. The estimated stiffness of the hair bundle was $k_{\mathrm{HB}} = 1.8 \pm 0.2\,\mathrm{mN \cdot m^{-1}}$ (mean ± SEM). As a result we can estimate the force exerted by radiation pressure that elicited the hair-bundle response shown in panel A to be approximately 40 pN. The hair bundle was bathed in perilymph for the whole duration of the experiment. The data point marked with a cross was not included in the fit. The stiffness of the flexible fiber $k_{\mathrm{p}} = 0.46\,\mathrm{mN \cdot m^{-1}}$ was measured by recording the Brownian motion of the fiber's tip in water.

