## [Decision Letter]

**Acceptance summary:**

The manuscript addresses the long-standing problem of engineering an in vitro stimulation method for individual inner ear sensory hair bundles that adequately provides a uniform and rapid stimulus characteristic of native inner ear stimulation. The authors address this unmet need with development and characterization of a light-based stimulus to generate rapid photonic force capable of deflecting a range of hair bundle geometries, including amphibian and mammalian vestibular and auditory hair bundles. The manuscript conveys a message that will be of use for the wide community of researchers working on mechanosensory integration and more broadly for engineers and scientists interested in using light to generate force. The study is extremely elegant, well written with beautiful illustrations. This work will be without a doubt a great addition to the field.

**Decision letter after peer review:**

Thank you for submitting your article "Rapid mechanical stimulation of inner-ear hair cells by photonic pressure" for consideration by *eLife*. Your article has been reviewed by 3 peer reviewers, including Fred Rieke as the Reviewing Editor and Reviewer #1, and the evaluation has been overseen by a Richard Aldrich as the Senior Editor. The following individual involved in review of your submission has agreed to reveal their identity: Claire Wyart (Reviewer #3).

As you will see, the reviewers were all enthusiastic about the approach you describe, but had some substantial concerns that need to be dealt with. In consultation, all three reviewers agreed about the importance of these points. First, additional evidence (beyond the spontaneous bundle movements) that the hair cells are viable after photonic stimulation is needed. Second, analysis of the robustness of the approach – i.e. how variable is it across cells and over time – is needed to evaluate potential applications. Third, the set of stimuli probed in the paper is quite limited. A better understanding of the range of stimuli that can be used is important for evaluating how useful the approach will be. These and some additional points are detailed in the individual reviews below.

Essential revisions:

*Reviewer #1 (Recommendations for the authors):*

The limitations of the approach could be included in the last paragraph of the introduction. It would similarly be useful in the discussion to not only compare photopic stimulation with other approaches, but to an ideal approach.

Is it possible to modulate the hair bundle position continuously - e.g. sinusoidally? If not, this would be useful to state as a limitation.

First paragraph of results. Could you elaborate a little here (a few additional sentences is probably enough)? The methods describes nicely why reflection alone is not sufficient, and some of the argument given there would demystify this paragraph.

Line 93: give approximate distance between fiber tip and hair bundle.

Lines 112-121: can you point out in the figure (e.g. with small arrows) the components of movement you describe here?

*Reviewer #2 (Recommendations for the authors):*

The manuscript addresses the long-standing problem of engineering a stimulation method for individual sensory hair cells in vitro that adequately provides a uniform and rapid stimulus characteristic of the native stimulus in the inner ear. The authors address this unmet need with development and characterization of a light-based stimulus to generate rapid photonic force capable of deflecting a range of hair bundle geometries, including amphibian and mammalian vestibular and auditory hair bundles. The writing is straightforward and easy to follow and figures are beautifully illustrated and informative. There are several shortcomings, attention to which, could further improve the manuscript and utility of the photonic stimulation method.

1) While the manuscript provides a significant technical advance, the end result does not necessarily inspire confidence that it can be widely implemented. For example, to be useful, the stimulator would need to provide a range of stimulus amplitudes to a single hair bundle. Likewise, a range of stimulus waveforms, steps, sinewaves of various frequencies, etc, would enhance the broad utility of the approach. Since the introduction section highlights the short comings of current hair bundle stimulation methods, it would also be of value for the results/Discussion section address whether the current photonic stimulation method has overcome those shortcomings or whether further technical development will be needed.

2) In general, the Results section is loosely quantified. For example, Figure 2A demonstrates significant cell-to-cell variability in the amplitude of the motion. What is the source of that variability? Biological variability in hair bundle stiffness, or variability in stimulus, probe position, light intensity, etc. Furthermore, what is the trial-to-trial variability for a single hair bundle? Figure 2 legend states each trace in panel 2A is an average of 25 responses, thus some representation of trial-to-trial variability could be quantified and presented. This would add value and provide the reader with a better sense of stimulus reproducibility.

3) A technical concern needs to be addressed to reassure readers that the photodiode signal is an accurate representation of hair bundle position. This has been well established in prior publications, but needs to be revisited here, either with additional experimentation or a sufficiently persuasive explanation. The concern is, since the stimulus is light itself and the response (bundle position) depends on a measurement of light signal, the stimulus could contaminate measurement of the response. This issue needs to be addressed in the Results section. If it’s buried in the methods section, I missed it, so please clarify.

4) The section entitled "Survival of mechanotransduction after laser irradiation" is important but somewhat unfulfilling. Measurement of spontaneous bundle motion is just one measure of intact mechanotransduction. It would be reassuring to know that other measures are also intact following hair bundle irradiation. Recordings of hair cell transduction current or receptor potentials, uptake of FM1-43, etc. could provide more direct evidence.

Line 112: References needed for "reported in the literature".

Line 125: How were tip links ruptured? Please clarify. Line 140: Stimuli were applied to 22 out har cells, but 13 bundles were defected? What happened to the other 9? No defection was evoked? Please clarify and explain.

*Reviewer #3 (Recommendations for the authors):*

There are only small modifications to be made to the manuscript in order to better characterize the variability of the responses induced in the hair bundle, a discussion on how the method could be used and validated in mammalian hair cells and a request to provide additional paths to check the viability of the cells and the robustness of the mechanosensory response after multiple optical stimulations have been performed.

1. The variability of the displacement to the 25 stimulations at 30mW @561nm in Figure 2A should be added as standard deviation (as a shade of light color) on top of the average depicted here. The variability in displacement for the rising as well as for the relaxation in B should also be depicted across stimulations for one cell and across cells.

Same indication of variability across trials and cells should apply for other figures where the average of 25 stimulations is depicted.

2. The authors make a point that mechanical stimulations are too slow to match the optimal frequency of activation of mammalian hair cells. However, if there is such variability in amplitude & kinetics of the displacement induced by the photonic force through the optic fiber, how can this technique be calibrated in small mammalian hair bundles?

3. The authors should check the viability of the cells and the robustness of the mechanosensory response after multiple optical stimulations have been performed. Currently they compare the spontaneous oscillations before and after a stimulation to illustrate that the method is not disrupting the function of the hair cell. However spontaneous oscillations are not visible on all cells. Are there other means (calcium imaging? electrophysiology?) by which the author could illustrate that the technique is not damaging the cell and altering the mechanosensory response in the hair bundle?

---

## [Author Response]

As you will see below, the reviewers were all enthusiastic about the approach you describe, but had some substantial concerns that need to be dealt with. In consultation, all three reviewers agreed about the importance of these points. First, additional evidence (beyond the spontaneous bundle movements) that the hair cells are viable after photonic stimulation is needed. Second, analysis of the robustness of the approach – i.e. how variable is it across cells and over time – is needed to evaluate potential applications. Third, the set of stimuli probed in the paper is quite limited. A better understanding of the range of stimuli that can be used is important for evaluating how useful the approach will be. These and some additional points are detailed in the individual reviews below.Reviewer #1 (Recommendations for the authors):The limitations of the approach could be included in the last paragraph of the introduction. It would similarly be useful in the discussion to not only compare photopic stimulation with other approaches, but to an ideal approach.

We have reviewed the main limitations in the Discussion section.

Is it possible to modulate the hair bundle position continuously - e.g. sinusoidally? If not, this would be useful to state as a limitation.

We have included a new illustration — Figure 4 — in which we show hair bundle responses to sinusoidal sweeps in frequencies between 10 Hz and 2000 Hz. We have also included in the Appendix a new figure (Figure 4) in which we show hair-bundle responses to continuously increasing and decreasing ramps. We have added a new section entitled “Variety of stimuli”:

“The fiber's power delivered onto a hair bundle can be modulated by changing the laser's power at the source. By combining analog and digital signals to drive the laser's output, we were able to stimulate hair bundles with an assortment of stimuli: sine waves, frequency sweeps, step pulses of various magnitudes, and continuously ascending and descending ramps (see Appendix 1, Figure 4). The responses of bullfrog's hair bundles to sinusoidal frequency sweeps at frequencies up to 2 kHz (Figure 4). In this case the upper boundary in the stimulus frequency was set by the ability of the hair bundle to follow, rather than by the limitations of the stimulation method.”

The caption of the new Figure 4 reads:

“Responses of hair bundles from the bullfrog's sacculus to sinusoidal frequency sweeps between 10 Hz and 200 Hz (A), 100 Hz and 500 Hz (B), and 1 kHz and 2 kHz (C). Each stimulus was achieved by driving the laser's source such that the amplitude of the sweep peaked at the maximum power output—12.5 mW for this fiber—while keeping its minimum above 0 mW. Each hair bundle was stimulated in the positive direction with 561 nm laser light; each trace is the average of 25 responses. Panel C portrays two 20 ms-long representative segments of the stimulus waveform, which would be unintelligible if displayed in full. These segments, located near the beginning and end of the sweep, are aligned with the magnification of the simultaneous hair-bundle response (red dashed boxes).

The caption of the new Figure 4 in Appendix 1 reads:

“Example of the variety of stimuli offered by photonic-force stimulation. (A) Frog hair bundles were stimulated in the positive direction with 561 nm light. The fiber's power output was increased in five steps from 0 mW to the maximum, 12.5 mW for this fiber. Each colored trace shown is the average of 25 responses tracking the hair bundle movement to a pulse of 50 ms at a constant laser power. (B) Frog's hair-bundle responses to increasing (left) and decreasing (right) ramps of 100 ms, in which the laser power was varied continuously between 0 mW and 12.5 mW.”

First paragraph of results. Could you elaborate a little here (a few additional sentences is probably enough)? The methods describes nicely why reflection alone is not sufficient, and some of the argument given there would demystify this paragraph.

We have modified the text: “Although an analysis based on reflection alone would indicate that a hair bundle is relatively insensitive to radiation pressure, geometric considerations reveal that multiple modes of light propagation occur in a hair bundle by virtue of the cylindrical shape of its stereocilia (see Material and methods). Each of these modes is capable of transferring momentum and therefore of mechanically stimulating the bundle.”

Line 93: give approximate distance between fiber tip and hair bundle.

We added “(typically about 7 μm)” in the text. Although this value is representative, the power output of the fiber, the orientation of the target hair bundle, and the narrowness of the laser conical beam ultimately determine the distance at which photonic force is most efficiently transmitted. This value must be determined heuristically.

Lines 112-121: can you point out in the figure (e.g. with small arrows) the components of movement you describe here?

We have added small arrows in Figures 2A and 2B and mentioned them in the text.

Reviewer #2 (Recommendations for the authors):The manuscript addresses the long-standing problem of engineering a stimulation method for individual sensory hair cells in vitro that adequately provides a uniform and rapid stimulus characteristic of the native stimulus in the inner ear. The authors address this unmet need with development and characterization of a light-based stimulus to generate rapid photonic force capable of deflecting a range of hair bundle geometries, including amphibian and mammalian vestibular and auditory hair bundles. The writing is straightforward and easy to follow and figures are beautifully illustrated and informative. There are several shortcomings, attention to which, could further improve the manuscript and utility of the photonic stimulation method.1) While the manuscript provides a significant technical advance, the end result does not necessarily inspire confidence that it can be widely implemented. For example, to be useful, the stimulator would need to provide a range of stimulus amplitudes to a single hair bundle. Likewise, a range of stimulus waveforms, steps, sinewaves of various frequencies, etc, would enhance the broad utility of the approach. Since the introduction section highlights the short comings of current hair bundle stimulation methods, it would also be of value for the results/Discussion section address whether the current photonic stimulation method has overcome those shortcomings or whether further technical development will be needed.

This comment is similar to that of Reviewer 1; please see our response above.

2) In general, the Results section is loosely quantified. For example, Figure 2A demonstrates significant cell-to-cell variability in the amplitude of the motion. What is the source of that variability? Biological variability in hair bundle stiffness, or variability in stimulus, probe position, light intensity, etc. Furthermore, what is the trial-to-trial variability for a single hair bundle? Figure 2 legend states each trace in panel 2A is an average of 25 responses, thus some representation of trial-to-trial variability could be quantified and presented. This would add value and provide the reader with a better sense of stimulus reproducibility.

We have added the 25 individual traces that constitute the average response displayed in Figure 2C. The traces are shown in gray and this is now mentioned in the caption.

3) A technical concern needs to be addressed to reassure readers that the photodiode signal is an accurate representation of hair bundle position. This has been well established in prior publications, but needs to be revisited here, either with additional experimentation or a sufficiently persuasive explanation. The concern is, since the stimulus is light itself and the response (bundle position) depends on a measurement of light signal, the stimulus could contaminate measurement of the response. This issue needs to be addressed in the Results section. If its buried in the methods section, I missed it, so please clarify.

The lights from the laser and LED light source had distinct wavelengths and were separated with filters and dichroic mirrors. Therefore, no laser light reached the photodiode. When the laser was pointed at a hair bundle adjacent to the one whose position was being tracked by the photodiode, no signal was detected by the photodiode, even though the laser's light brightened the entire field of view.

4) The section entitled "Survival of mechanotransduction after laser irradiation" is important but somewhat unfulfilling. Measurement of spontaneous bundle motion is just one measure of intact mechanotransduction. It would be reassuring to know that other measures are also intact following hair bundle irradiation. Recordings of hair cell transduction current or receptor potentials, uptake of FM1-43, etc. could provide more direct evidence.

The fact that hair bundles continued to oscillate spontaneously after irradiation means that not only the mechanotransduction apparatus was not damaged but, more subtly, that the hair bundles continued to reside in the same region of the phase space—meaning that the control parameter governing the transition to the oscillatory regime was not perturbed by irradiation. In Appendix 1—figure 6, we have now added results showing the persistent uptake of FM1-43 by hair bundles after extensive irradiation.

We have added a paragraph to the section “Survival of mechanotransduction after laser irradiation”:

“To further assess the health of the hair bundles exposed to laser irradiation, we compared their intake of FM1-43—a fluorescent dye that enters a hair cell through open mechanotransduction channels (Gale *et al.*, 2001)—with that of surrounding undisturbed hair cells and that of a mechanically damaged bundle (Appendix 1—figure 6). The fluorescence signal from laser-irradiated hair bundles showed no visible difference with respect to those of the unscathed cells, whereas mechanically damaged bundles were visibly dimmer. The diminished fluorescence likely resulted from breakage of tip links that reduced the opening of the mechanotransduction channels, thereby limiting the intake of the dye.”

The caption of the new Appendix 1—figure 6 reads:

“The fluorescence signal of bullfrog saccular hair bundles loaded with FM1-43 after exposure to laser irradiation. We stimulated successive hair bundles in a row with 50 ms pulses at a power of 12.5 mW, the full power available for the fiber. The five hair bundles in the green box were subjected to 10, 20, 30, 40, and 50 pulses, as indicated. A sixth hair bundle in the same row, labeled in red, was damaged by being crushed a few times by the end of the optical fiber. Immediately after those procedures, the sample was exposed to 1 mM of FM1-43 for one minute, after which fluorescence was recorded. The stimulated hair bundles show comparable levels of brightness between each other and with respect to those in the surroundings. The loading of the dye is visibly reduced in the mechanically damaged cell owing to breakage of the tip links.”

Line 112: References needed for "reported in the literature".

We have added references.

Line 125: How were tip links ruptured? Please clarify.

The tip links were ruptured by exposure to the Ca^2+^ chelator BAPTA. We have now mentioned this in the text and in the caption of the figure.

Line 140: Stimuli were applied to 22 out har cells, but 13 bundles were defected? What happened to the other 9? No defection was evoked? Please clarify and explain.

Although we are not sure why those nine hair bundles did not move, we note that the hair bundle of an outer hair cell forms a triangular prism that can produce forward scattering for oblique illumination. Forward scattering might create a negative force; indeed, this phenomenon has been exploited in optical tractor beams (Sukhov and Dogariu, Phys. Rev. Lett., 2011). This negative force will add to the positive force due to back scattering. The relative contribution of the two forces is in general a complex function of the object’s geometry and optical properties as well as of the beam’s profile and angle of incidence. It is therefore possible, and indeed expected, that some hair bundles move more than others in the direction of beam propagation.

Reviewer #3 (Recommendations for the authors):There are only small modifications to be made to the manuscript in order to better characterize the variability of the responses induced in the hair bundle, a discussion on how the method could be used and validated in mammalian hair cells and a request to provide additional paths to check the viability of the cells and the robustness of the mechanosensory response after multiple optical stimulations have been performed.1. The variability of the displacement to the 25 stimulations at 30mW @561nm in Figure 2A should be added as standard deviation (as a shade of light color) on top of the average depicted here. The variability in displacement for the rising as well as for the relaxation in B should also be depicted across stimulations for one cell and across cells.

This comment is similar to that of Reviewer 2; as noted above, we have added 25 individual traces to illustrate the small variability.

Same indication of variability across trials and cells should apply for other figures where the average of 25 stimulations is depicted.

Showing individual traces for all the figures would clutter them extensively. For each recording condition, the variability of responses across trials was always small. Because we measured displacements in the nanometer range, the principal purpose of repeated stimulation was to average away the noise owing to thermal agitation of the bundles and to photon shot noise in the photodiode system. Please also see our response to the related comment below.

2. The authors make a point that mechanical stimulations are too slow to match the optimal frequency of activation of mammalian hair cells. However, if there is such variability in amplitude & kinetics of the displacement induced by the photonic force through the optic fiber, how can this technique be calibrated in small mammalian hair bundles?

The variability is inevitable because the method applies force and not displacement. Therefore, the motion of each hair bundle is determined by its stiffness and drag coefficient (and, at higher frequencies, its mass). Because hair-bundle stiffness is known to vary – and can even become negative owing to the phenomenon of gating compliance –this variability is expected, and indeed a welcome indication of healthy bundles. Given the rich panoply of known active hair-bundle behaviors, it would indeed be strange if all hair bundles moved identically in response to a force step.

We discuss the issue of calibration at the end of the Discussion. In addition, we have added Figure 7 of Appendix 1, in which we determine the stiffness of an individual hair bundle with a calibrated glass fiber, then ascertain what photonic stimulus yields a similar deflection. Although in principle a bundle's stiffness can be also deduced from its Brownian motion, mammalian hair bundles are relatively stiff and it is difficult to measure their Brownian motion without laser interferometry.

Perhaps most importantly, one needs to know the stimulus force exactly when one wishes to measure hair-bundle stiffness accurately. But the stiffness of bundles is already well known; for example, hair bundles of outer hair cell have stiffness values that vary between about 1 mN·m^–1^ and 4 mN·m^–1^. So if such a hair bundle – for example the one illustrated in Figure 3 – was deflected by 10 nm with photonic force, the associated force was 10–40 pN. The precise force is not critical in estimates of the time constant of hair-bundle deflection.

Finally, stiffness can be estimated by applying forces using slower traditional methods. Our method is designed to apply whatever force it takes to rapidly move a hair bundle by a desired and directly measured distance.

3. The authors should check the viability of the cells and the robustness of the mechanosensory response after multiple optical stimulations have been performed. Currently they compare the spontaneous oscillations before and after a stimulation to illustrate that the method is not disrupting the function of the hair cell. However spontaneous oscillations are not visible on all cells. Are there other means (calcium imaging? electrophysiology?) by which the author could illustrate that the technique is not damaging the cell and altering the mechanosensory response in the hair bundle?

We concur with this suggestion, and have performed an additional control experiment. Kindly see our response to comment 4 of Reviewer 1.